# Flexible categorization in perceptual decision making

Genís Prat-Ortega [1,2✉], Klaus Wimmer [2,3,4], Alex Roxin[2,3,4,5] & Jaime de la Rocha [1,4,5✉]

Perceptual decisions rely on accumulating sensory evidence. This computation has been studied using either drift diffusion models or neurobiological network models exhibiting winner-take-all attractor dynamics. Although both models can account for a large amount of data, it remains unclear whether their dynamics are qualitatively equivalent. Here we show that in the attractor model, but not in the drift diffusion model, an increase in the stimulus fluctuations or the stimulus duration promotes transitions between decision states. The increase in the number of transitions leads to a crossover between weighting mostly early evidence (primacy) to weighting late evidence (recency), a prediction we validate with psychophysical data. Between these two limiting cases, we found a novel *flexible categorization* regime, in which fluctuations can reverse initially-incorrect categorizations. This reversal asymmetry results in a non-monotonic psychometric curve, a distinctive feature of the attractor model. Our findings point to correcting decision reversals as an important feature of perceptual decision making.

[1] Institut d'Investigacions Biomèdiques August Pi i Sunyer (IDIBAPS), Barcelona 08036, Spain. [2] Centre de Recerca Matemàtica (CRM), Campus de Bellaterra, Edifici C, 08193 Bellaterra, Barcelona, Spain. [3] Barcelona Graduate School of Mathematics, Barcelona, Spain. [4] These authors jointly supervised this work: Klaus Wimmer, Alex Roxin, Jaime de la Rocha. [5] These authors contributed equally: Alex Roxin, Jaime de la Rocha. ✉email: genisprat@gmail.com; jrochav@clinic.cat

Integrating information over time is a fundamental computation that neural systems can adaptively perform in a variety of contexts. The integration of perceptual evidence is an example of such computation, and its most common paradigm is the binary categorization of ambiguous stimuli characterized by a stream of sensory evidence. This process is typically modeled with the drift diffusion model with absorbing bounds (DDMA) which integrates the stimulus evidence linearly until one of the bounds is reached[1]. The DDMA and its different variations have been successfully used to fit psychometric and chronometric curves[1,2], to capture the speed-accuracy trade-off[1–3], to account for history dependent choice biases[4], changes of mind[5], confidence reports[6], or the Weber's law[7]. Although the absorbing bounds were originally thought of as a mechanism to terminate the integration process, the DDMA has also been applied to fixed duration tasks[8–10]. In motion discrimination tasks, for instance, it can reproduce the subjects' tendency to give more weight to early rather than late stimulus information, which is called a primacy effect[8,10–14]. However, depending on the details of the task and the stimulus, subjects can also give more weight to late rather than to early evidence (i.e., a recency effect)[15,16] or weigh the whole stimulus uniformly[17]. In order to account for these differences, the DDMA needs to be modified by using reflecting instead of absorbing bounds or by removing the bounds altogether[18]. Despite their considerable success in fitting experimental data, the DDMA and its many variants remain purely phenomenological descriptions of sensory integration. This makes it difficult to link the DDMA to the actual neural circuit mechanisms underlying perceptual decision making.

These neural circuit mechanisms have been studied with biophysical attractor network models that can integrate stimulus evidence over relatively long time scales[19,20]. Attractor network models have been used, among other examples, to study the adjustment of speed-accuracy trade-off in a cortico-basal ganglia circuit[21], learning dynamics of sensorimotor associations[22], the generation of choice correlated sensory activity in hierarchical networks[23–25], the role of the pulvino-cortical pathway in controlling the effective connectivity within and across cortical regions[26] or how trial history biases can emerge from the circuit dynamics[27]. In the typical regime in which the attractor network was originally used for perceptual categorization[19,28], the impact of the stimulus on the decision decreases as the network evolves towards an attractor. In this regime, the integration dynamics of the attractor model are qualitatively similar to those of the DDMA as it also shows a primacy effect. Moreover, the attractor network can also provide an excellent fit to psychometric and chronometric curves[19,28]. Thus, a common implicit assumption is that the attractor network is simply a neurobiological implementation of the DDMA[29,30] and hence there has been more interest in studying the similarities between these two models rather than their differences[31] (but see refs. [32,33]).

Here, we show that the attractor model has richer dynamics beyond the well known primacy regime. In particular, the model exhibits a crossover from primacy to recency as the stimulus fluctuations or stimulus duration are increased. Intermediate to these two limiting regimes, the stimulus can impact the upcoming decision nearly uniformly across the entire stimulus duration. Specifically, if the first attractor state reached corresponds to the incorrect choice, stimulus fluctuations later in the trial can lead to a correcting transition, while if the initial attractor is correct, fluctuations are not strong enough to drive an error transition. As a consequence, the model shows a non-monotonic psychometric curve as a function of the strength of stimulus fluctuations, and the maximum occurs precisely in this intermediate "flexible categorization" regime. To illustrate the relevance of our theoretical results, we re-analyze data from two psychophysical experiments[34,35] and show that the attractor model can quantitatively fit the crossover from primacy to recency with the stimulus duration, and the integration and storage of evidence when stimuli are separated by a memory delay. Our characterization of the flexible categorization regime in the attractor model reveals that correcting transitions may be a key property of evidence integration in perceptual decision making.

## Results

**Canonical models of perceptual decision making result in stereotypical psychophysical kernels.** We start by characterizing the dynamics of evidence integration in standard drift diffusion models during a binary classification task. These models are described as the evolution of a decision variable $x(t)$ that integrates the moment-by-moment evidence $S(t)$ provided by the stimulus, plus a noise term reflecting the internal stochasticity in the process[1,30,31].

$$\tau \frac{dx}{dt} = S(t) + \sigma_I \xi_I(t), \qquad (1)$$

where $\tau$ is the time constant of the integration process. The evidence $S(t)$ fluctuates in time and can be written as a constant mean drift $\mu$, plus a time-varying term, caused by the fluctuations of the input stimulus: $S(t) = \mu + \sigma_S \xi_S(t)$. We call $\sigma_S$ the magnitude of stimulus fluctuations. Assuming that both fluctuating terms, $\xi_I$ and $\xi_S$ are Gaussian stochastic processes, Eq. 1 can be recast as the motion of a particle in a potential:

$$\tau \frac{dx}{dt} = -\frac{d\varphi(x)}{dx} + \sigma_S \xi_S(t) + \sigma_I \xi_I(t), \qquad (2)$$

where the potential $\varphi(x) = -\mu x$ (Fig. 1d–f, inset). The conceptual advantage of using a potential relies on the fact that the dynamics of the decision variable always "roll downward" towards the minima of the potential with only the fluctuation terms $\xi_S(t)$ or $\xi_I(t)$ causing possible motion upward. Notice that, although the term $\xi_S(t)$ is modeled as a noise term, it represents the temporal variations of the stimulus which are under the control of the experimenter. The existence of decision bounds can be readily introduced in the shape of the potential, which strongly influences how stimulus fluctuations impact the upcoming decision. To quantify this impact, we used the Psychophysical Kernel (PK) which measures the influence of the stimulus fluctuations on the decision during the course of the stimulus (see Methods): (1) In the DDMA (Fig. 1a), absorbing bounds are implemented as two vertical "cliffs" such that when the decision variable arrives at one of them, it remains there for the rest of the trial. When this happens, the fluctuations late in the stimulus are unlikely to affect the decision, yielding a decaying PK characteristic of a "primacy" effect[8,18,31,33,36]. (2) In the Drift Diffusion Model with Reflecting bounds (DDMR) (Fig. 1b), the bounds are two vertical walls that set limits to the accumulated evidence; early stimulus fluctuations are largely forgotten once the decision variable bounces against one bound and hence the PK shows a "recency" effect[18]. (3) In the Perfect Integrator (PI) (Fig. 1c), there are no bounds, the stimulus is integrated uniformly across time yielding a flat PK[17]. Thus, each of these three *canonical* models performs a qualitatively distinct integration process by virtue of how the bounds are imposed. Moreover, the characteristic integration dynamics of each model is invariant to changes in the stimulus parameters. To illustrate this, we show how the PKs depended on the magnitude of the stimulus fluctuations, $\sigma_S$ (Fig. 1). For very weak stimulus fluctuations, all three models are trivially equivalent because the bounds are never reached and hence the PKs are flat (Fig. 1d–f). As $\sigma_S$ increases, in both the DDMA and the DDMR, the bounds are reached faster yielding an increase and a decrease of the PK slope, respectively (Fig. 1h). In these two models, the integration

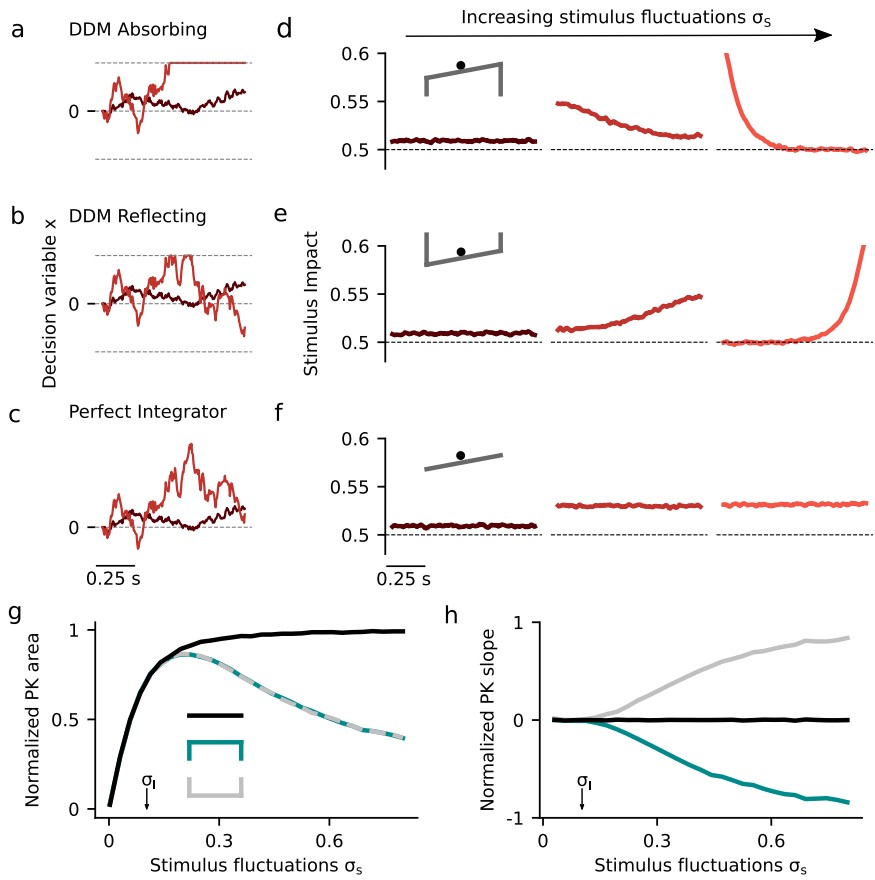

**Fig. 1 Dynamics of evidence accumulation in the canonical drift diffusion models. a–c** Single-trial example traces of the decision variable $x(t)$ for weak ($\sigma_S = 0.03$) and intermediate ($\sigma_S = 0.28$) stimulus fluctuations in the three canonical models. **a** The DDM with absorbing bounds integrates the stimulus until it reaches one of the absorbing bounds represented in the potential landscape as infinitely deep attractors (see inset in **d**). The slope of the potential landscape is the mean stimulus strength, in this case $\mu < 0$. **b** The DDM with reflecting bounds integrates the stimulus linearly until a bound is reached when no more evidence can be accumulated in favor of the corresponding choice option (see inset in **e**). **c** The perfect integrator integrates the entire stimulus uniformly, corresponding to a diffusion process with a flat potential (see inset in **f**). In the three models, the choice is given by the sign of $x(t)$ at stimulus offset. **d–f** Psychophysical Kernels (PK) for the three canonical models for increasing magnitude of the stimulus fluctuations (from left to right): $\sigma_S = 0.03$, 0.28, and 0.69. The PK measures the time-resolved impact of the stimulus fluctuations on choice (see Methods). **g–h** Normalized PK area and normalized PK slope as a function of $\sigma_S$ for the three canonical models (see inset in **g** for color code). The area is normalized by the PK area of the perfect integrator with no internal noise ($\sigma_i = 0$) and hence measures the ability of each model to integrate the stimulus fluctuations. In all panels, internal noise was fixed at $\sigma_i = 0.1$ (see arrows in **g** and **h**) which was sufficiently small to prevent $x(t)$ from reaching the bounds in the absence of a stimulus. Mean stimulus evidence was $\mu = 0$ in all cases.

of evidence becomes more and more transient as $\sigma_S$ increases, ultimately causing a decrease of the PK area (Fig. 1g). Including collapsing bounds in the DDMA did not modify the picture qualitatively, with the integration becoming more transient as the velocity of the collapsing bounds increased (Supplementary Fig. 1). The PK for the PI remains flat for all $\sigma_S$ (zero PK slope, Fig. 1h) and its area increases monotonically (Fig. 1g). Thus, the dynamics of evidence accumulation are an invariant and distinct property of each model.

**Neurobiological models show a variety of integration regimes**. We next characterized the dynamics of evidence accumulation in the double well model (DWM), which can accurately describe the dynamics of a biophysical attractor network model[19,28]. The DWM exhibits winner-take-all attractor dynamics defined by the nonlinear potential $\varphi(x)$:

$$\varphi(x) = -\mu x - \alpha x^2 + x^4. \qquad (3)$$

The resulting energy landscape can exhibit two minima (i.e., attractor states) corresponding to the two possible choices

(Fig. 2a, inset). The three terms of the potential, from left to right, capture (1) the impact of the net stimulus evidence $\mu$ which, as in the canonical models, tilts the potential towards the attractor associated with the correct choice; (2) the model's internal categorization dynamics parameterized by the height of the barrier separating the two attractors (which scales with $\alpha^2$), and (3) bounds, also arising from the internal dynamics, that limit the range over which evidence is accumulated. We found that the DWM had a much richer dynamical repertoire as a function of stimulus fluctuations magnitude than the canonical models. Specifically, the attractors imposed the categorization dynamics, but these could be overcome by sufficiently strong stimulus fluctuations. Thus, for weak $\sigma_S$, the categorization dynamics dominated: when the system reached an attractor, it remained in this initial categorization until the end of the stimulus. In this regime, only early stimulus fluctuations occurring before reaching an attractor could influence the final choice, yielding a primacy PK[19,23] (Fig. 2c, second line from the left). For strong $\sigma_S$, the initial categorization had no impact on the final choice because transitions between the attractors occurred readily. It was the fluctuations coming late in the trial which determined the final

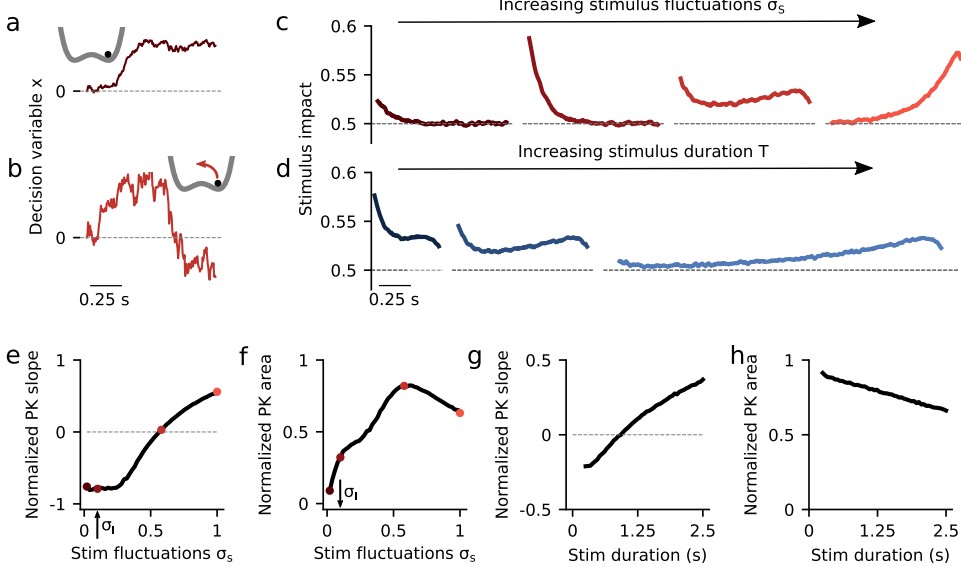

**Fig. 2 Dynamics of evidence accumulation in the double well model. a**, **b** Single-trial example traces of the decision variable for the DWM with weak (**a**, $\sigma_S = 0.1$) and intermediate (**b**, $\sigma_S = 0.58$) stimulus fluctuations $\sigma_S$. Transitions between attractors were only possible for sufficiently strong $\sigma_S$ (insets). **c** PKs for increasing values of $\sigma_S$ (from left to right, $\sigma_S = 0.02$, 0.1, 0.58, and 1). **d** PKs for increasing values of stimulus duration $T$ (from left to right, $T = 0.5$, 1, and 2.5 with $\sigma_S = 0.58$). **e**, **f** Normalized PK slope and PK area as a function of $\sigma_S$. Colored dots indicate the examples shown in panel c. The area peaks at the flexible categorization and it vanishes for small $\sigma_S$ because choice is then driven by internal noise. **g**, **h** Normalized PK slope and area as a function of $T$ with $\sigma_S = 0.58$. As $T$ increases, the DWM integrates a smaller fraction of the stimulus making the area decrease monotonically. Internal noise was $\sigma_I = 0.1$ in all panels (see arrows in panels **e** and **f**).

state of the system and hence the PK showed recency (Fig. 2c, orange). For moderate values of $\sigma_S$, there was an intermediate regime in which the PK was a mixture between primacy and recency, but not necessarily flat (Fig. 2c, third line from the left). We called this regime *flexible categorization* because it represented a balance between the internal categorization dynamics and the ability of the stimulus fluctuations to overcome their attraction (Fig. 2b). As a result of this balance, the stimulus fluctuations impacted the choice over the whole trial (PK slope = 0; Fig. 2e) because both initial fluctuations and later fluctuations causing transitions had a substantial impact on choice. Moreover, these fluctuations causing transitions were more temporally extended than those in the recency regime (Supplementary Fig. 2a). Thus, the area of the PK reached its maximum (maximum area = 0.82; Fig. 2f) implying that the integration of the stimulus fluctuations carried out by DWM was comparable to a PI (which has PK area equal 1). The same crossover from primacy to recency regimes, passing through the flexible categorization regime, could be achieved, at fixed $\sigma_S$, by varying the stimulus duration (Fig. 2d, g). This occurs because for a fixed magnitude of stimulus fluctuations, the *rate* of transitions was constant but the probability to observe a transition increased with the stimulus duration changing the shape of the PK accordingly (Fig. 2d). In sum, depending on the capacity of the stimulus to generate transitions between attractors, the DWM model could operate in the primacy, the recency, or the flexible categorization integration regime.

**Decision accuracy in models of evidence integration.** Given that the DWM changes its integration regime when $\sigma_S$ is varied, we next investigated the impact of this manipulation on the decision accuracy. We set the internal noise to $\sigma_I = 0$ and computed the psychometric function $P(\mu, \sigma_S)$ showing the proportion of correct choices as a function of the mean stimulus evidence $\mu$ and the strength of stimulus fluctuations $\sigma_S$. For small fixed $\sigma_S$ the section of this surface yielded a classic sigmoid-like psychometric curve

$P(\mu)$ (Fig. 3a, dark brown curve). As $\sigma_S$ increased, this curve became shallower simply because larger fluctuations implied a drop in the signal-to-noise ratio of the stimulus (Fig. 3a, red and orange curves). Unexpectedly, however, the decline in sensitivity of the curve $P(\mu)$ was non-monotonic (Fig. 3a), an effect which was best illustrated by plotting the less conventional psychometric curve $P(\sigma_S)$ at fixed $\mu$ (Fig. 3a, b, black curve). To understand this non-monotonic dependence, we first defined two transition probabilities: the *correcting* transition probability $p_C$ was the probability to be in the correct attractor at the end of a trial, given that the first visited attractor was the error. The *error-generating* transition probability $p_E$ was the opposite, i.e., the probability to finish in the wrong attractor given that the correct one was visited first (see Methods). Using Kramers' reaction-rate theory[37] the transition probabilities could be analytically computed, and the accuracy $P$ could be expressed as the probability to initially make a correct categorization and maintain it, plus the probability to make an initial error and reverse it:

$$P = P_0(1 - p_E) + (1 - P_0)p_C, \quad (4)$$

where $P_0$ was the probability of first visiting the correct attractor (Methods). When the fluctuations were negligible $\sigma_S \approx 0$, the decision variable always rolled down towards the correct choice because the double well potential was tilted to the correct attractor (e.g., $\mu > 0$), and hence $P = 1$ (Fig. 3b *i*). As $\sigma_S$ started to increase, fluctuations early in the stimulus could cause the system to fall into the incorrect attractor but, because fluctuations were not sufficiently strong to generate transitions ($p_E \approx p_C \approx 0$), accuracy was $P = P_0$ (Eq. 4) and decreased with $\sigma_S$ towards chance (gray line in Fig. 3b). As the stimulus fluctuations grew stronger, the transitions between attractors became more likely but, because the barrier to escape from the incorrect attractor was smaller than the barrier to escape from the correct attractor, the two transition probabilities were very different. Specifically, Kramers' theory shows that the ratio between $p_C$ and $p_E$ depends exponentially on the barrier height difference (see Methods). Thus, $p_C$ increased steeply with $\sigma_S$, even before $p_E$ reached non-

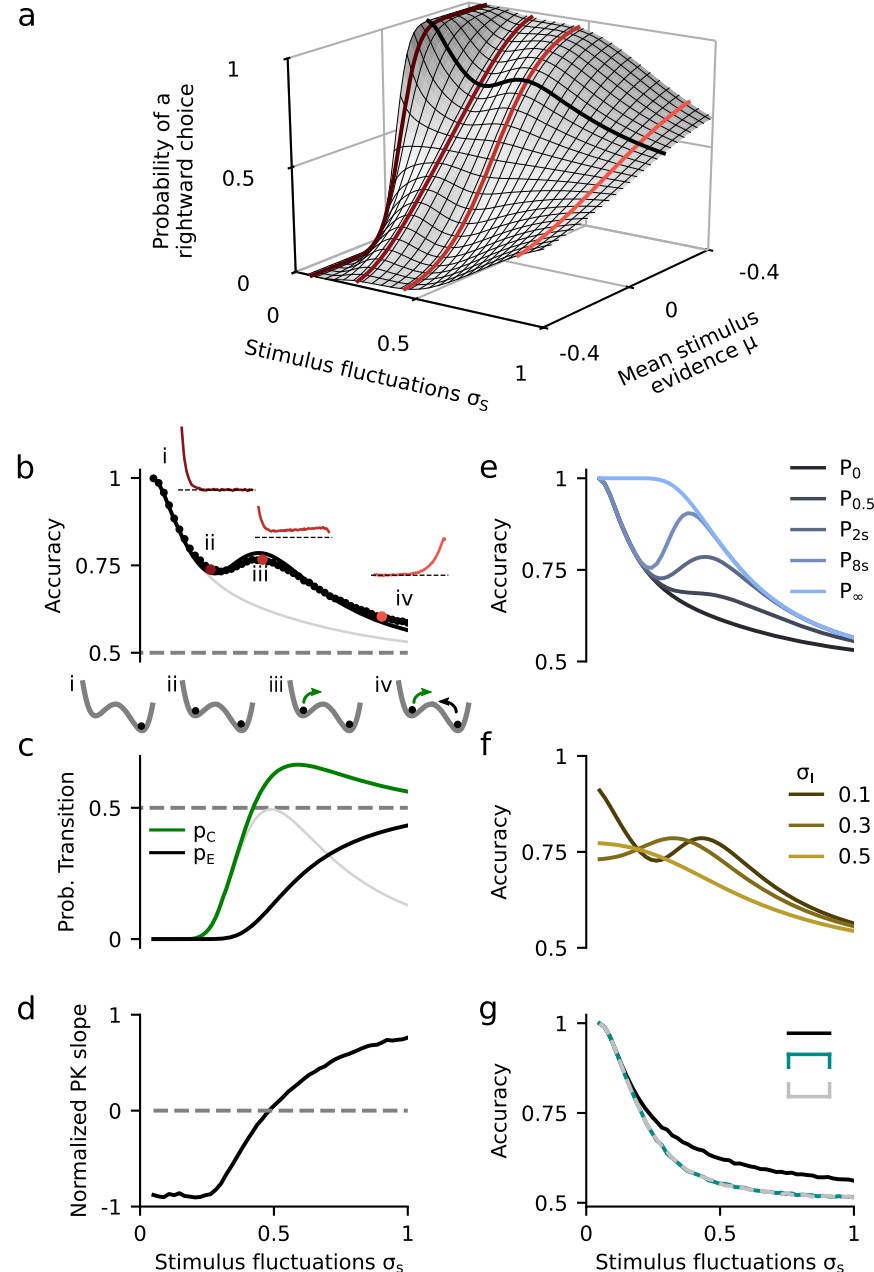

**Fig. 3 Impact of stimulus fluctuations on choice accuracy in the double well model. a** Probability of a rigward choice as a function of the mean stimulus evidence ($\mu$) and the stimulus fluctuations ($\sigma_S$). The colored lines show *classic* psychometric curves, accuracy versus $\mu$ (for fixed $\sigma_S = 0.07$, 0.26, 0.46, and 0.90) whereas the black line shows the accuracy versus $\sigma_S$ (for fixed $\mu = 0.15$). **b** Accuracy ($P$) as a function of the stimulus fluctuations $\sigma_S$ obtained from numerical simulations (dots) and theory (line, same as black line in **a**). Insets show the PK for three values of $\sigma_S$ (marked with colored dots). The gray line shows the accuracy of the first visit attractor ($P_0$). **c** Probability to make a correcting $p_C$ (green) or an error transition $p_E$ (black) and their difference $p_C - p_E$ (gray). The local maximum in $P$ coincides with the maximum difference between the two probabilities. Insets: sequence of regimes as transitions become more likely: (i) For negligible $\sigma_S$, the decision variable always evolves towards the correct attractor; (ii) as $\sigma_S$ increases, the decision variable can visit the incorrect attractor but neither kind of transition is activated; (iii) for stronger $\sigma_S$, only the correcting transitions (green arrow) are activated; (iv) for strong $\sigma_S$, both types of transition are activated. **d** Normalized PK slope as a function of $\sigma_S$. The flexible categorization regime, reached when the index is close to zero, coincides with the local maximum in accuracy (**a**). **e** Accuracy versus $\sigma_S$ for different stimulus durations $T$ (see inset). The accuracy for any finite $T$ shifts as $\sigma_S$ increases between the probability to first visit the correct attractor $P_0$ and the stationary accuracy $P_\infty$. **f** Accuracy versus $\sigma_S$ for different magnitudes of the internal noise (see inset). **g** Accuracy versus $\sigma_S$ for the three canonical models (see inset). The internal noise was $\sigma_i = 0$ in all panels except in **f**.

negligible values (Fig. 3c) opening a window in which transitions were *only* correcting: accuracy became $P \simeq P_0 + (1 - P_0)p_C$ and it increased with $\sigma_S$ (Fig. 3b *iii*). The maximum difference between $p_C$ and $p_E$ coincided with the flexible categorization regime in which the PK slope was zero and the accuracy showed a

local maximum (Fig. 3b–d). Finally, for strong $\sigma_S$, error transitions also became likely and the net effect of stimulus fluctuations was again deleterious, causing a decrease of $P$. In sum, it was the large difference in transition probabilities caused by the double well landscape which led to the non-monotonic dependence of

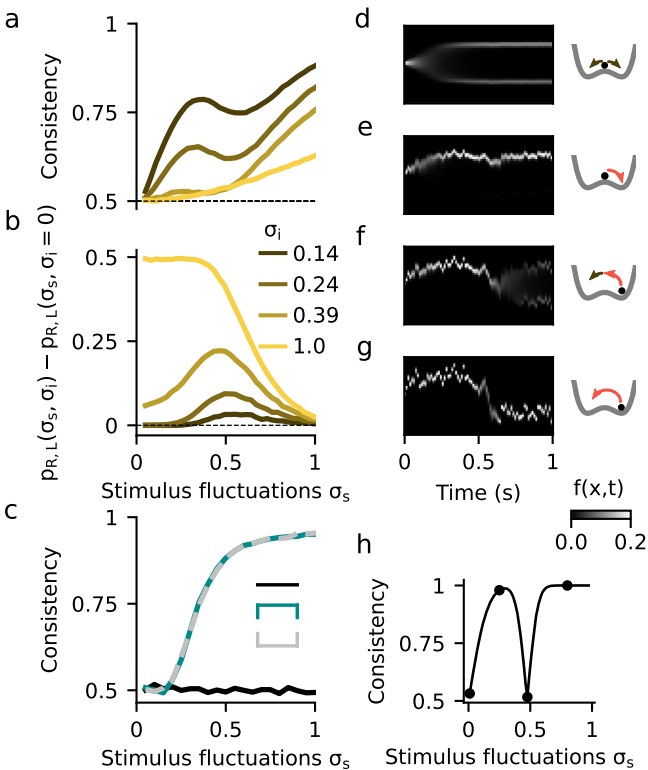

**Fig. 4 Dependence of choice consistency on stimulus fluctuations.**
**a** Average consistency versus stimulus fluctuations $\sigma_S$ for different values of the internal noise $\sigma_I$ (see inset in **b**). **b** Difference between the transition probabilities with ($p_{R,L}(\sigma_S, \sigma_I)$) and without ($p_{R,L}(\sigma_S, \sigma_I = 0)$) internal noise. The drop in consistency coincides with an increase of this difference revealing the $\sigma_S$-range in which transitions occurred because of the cooperation of internal and stimulus fluctuations. **c** Consistency versus $\sigma_S$ for the canonical models. The consistency of the perfect integration is at chance level because we used stimuli with exactly zero integrated evidence (see Methods). **d–g** Temporal evolution of the decision variable probability distribution $f(x,t)$ for an example stimulus in the different regimes of $\sigma_S$: for negligible $\sigma_S$ the choice is driven by the internal noise and the consistency is very low (53.2%, **d**). For small $\sigma_S$, when the stimulus determines the first visited attractor but fluctuations are not strong enough to produce transitions, the consistency is very high (97.8%, **e**). For intermediate $\sigma_S$, the transitions can only occur when $\sigma_I$ and $\sigma_S$ work together to cause a large fluctuation. Because the internal noise has again impact on the choice, the consistency decreases (51.7%, **f**). For large $\sigma_S$, the stimulus fluctuations are strong enough to produce transitions by itself and the consistency is again very high (100%, **g**). **h** Consistency versus $\sigma_S$ obtained just using the example stimulus shown in **d–g** (points mark the $\sigma_S$ values shown in **d–g**). Mean stimulus evidence was $\mu = 0$ in all panels.

the psychometric curve. Because the canonical models lacked attractor dynamics, the accuracy in all of them decayed monotonically with the stimulus fluctuations (Fig. 3g).

We next asked whether the non-monotonicity of the psychometric curve was robust to variation of other parameters such as the mean stimulus evidence $\mu$, the stimulus duration $T$, and the internal noise $\sigma_I$. We found that the non-monotonicity was robustly obtained over a broad range of $\mu$, ranging from small values just above zero to a critical value beyond which the curve became monotonically decreasing (Supplementary Fig. 3). Because the transition probabilities scale with the stimulus duration $T$, the psychometric curve $P(\sigma_S)$ was strongly affected by changes in $T$ (Fig. 3e). To understand this dependence, we rewrote the transitions probabilities $p_C$ and $p_E$ from Eq. 4 as a

function of the transition rates and the stimulus duration (see Methods, Eqs. 17 and 18):

$$P = P_0 exp(-kT) + P_\infty[1 - exp(-kT)], \qquad (5)$$

where $k$ is the sum of the transition rates from both attractors and $P_\infty$ is the stationary accuracy (i.e., the limit of $P$ when $T\rightarrow\infty$). As expected, the two psychometric curves $P_0$ and $P_\infty$, which decreased monotonically with $\sigma_S$, delimited the region in which $P$ existed: for weak $\sigma_S$, $P$ followed the decay of the psychometric curve $P_0$, whereas for strong $\sigma_S$ it tracked the decay of the stationary accuracy $P_\infty$. The switching point occurred when the probability to observe a transition was substantial, i.e., when $kT \sim 1$. For longer stimulus durations, the activation of the transitions occurred for weaker $\sigma_S$ and consequently the bump in accuracy was shifted towards the left also becoming more prominent (Fig. 3e, Methods). For very short $T$, the activation of the transitions occurred for such a large value of $\sigma_S$ that the two curves $P_0$ and $P_\infty$ have come too close and the psychometric $P(\sigma_S)$ was then monotonically decreasing (Fig. 3e). Finally, when we set the internal noise to a nonzero value, it sets a minimal level of fluctuations below which no stimulus magnitude $\sigma_S$ could go, effectively cropping the psychometric curve $P(\sigma_S)$ from the left (Fig. 3f). Only when the internal noise was larger than a critical value the psychometric curve became monotonically decreasing (Supplementary Fig. 3, see Methods for the computation of the critical noise value). In sum, the non-monotonicity of the psychometric curve was a robust effect, being most prominent for values of the mean stimulus evidence $\mu$ yielding an intermediate accuracy (i.e., $P \sim 0.75$), long stimulus durations and weak internal noise.

**Consistency in models of evidence integration.** In order to identify further signatures of the nonlinear attractor dynamics that could be tested experimentally, we studied the choice consistency of the DWM. Choice consistency is defined as the probability that two presentations of the same exact stimulus, i.e., the same realization of the stimulus fluctuations, yield the same choice. In the absence of internal noise, the decision process in the model is deterministic and consistency is 1. In contrast, when the stimulus has no impact on the choice, the consistency is 0.5. We used the double-pass method, which presents each stimulus twice[12,38,39], to explore how consistency in the DWM depended on $\sigma_S$ and $\sigma_I$ (Fig. 4). We only used $\mu = 0$ stimuli with exactly zero integrated evidence in order to avoid the parsimonious increase of consistency due to larger deviations of the accumulated evidence from the mean (see Methods). As expected, consistency was close to 0.5 when $\sigma_S$ was small compared to $\sigma_I$, and it increased with increasing $\sigma_S$ (Fig. 4a). However, despite this general increase, we found a striking drop in consistency for a range of intermediate $\sigma_S$ values. Thus, consistency could depend non-monotonically on the strength of stimulus fluctuations, a similar effect as observed for choice accuracy. To understand this effect, we studied the time-course of the decision variable $x$ over many repetitions of a single stimulus, at different values of $\sigma_S$ (Fig. 4d–h). For very small $\sigma_S$, consistency was 0.5 because the internal noise was the dominant factor making both choices equally likely (Fig. 4d). As $\sigma_S$ grew, stimulus fluctuations could determine the first visited attractor but decision reversals were still not activated, yielding a high consistency (Fig. 4e). For larger $\sigma_S$, transitions occurred but only when internal noise and the stimulus fluctuations worked together to produce a large fluctuation (Fig. 4f). The necessary contribution of the internal noise, that varied from trial-to-trial, led to the decrease in consistency. Once $\sigma_S$ was large enough to cause reversals on its own, consistency increased again (Fig. 4g). Thus, as with the non-

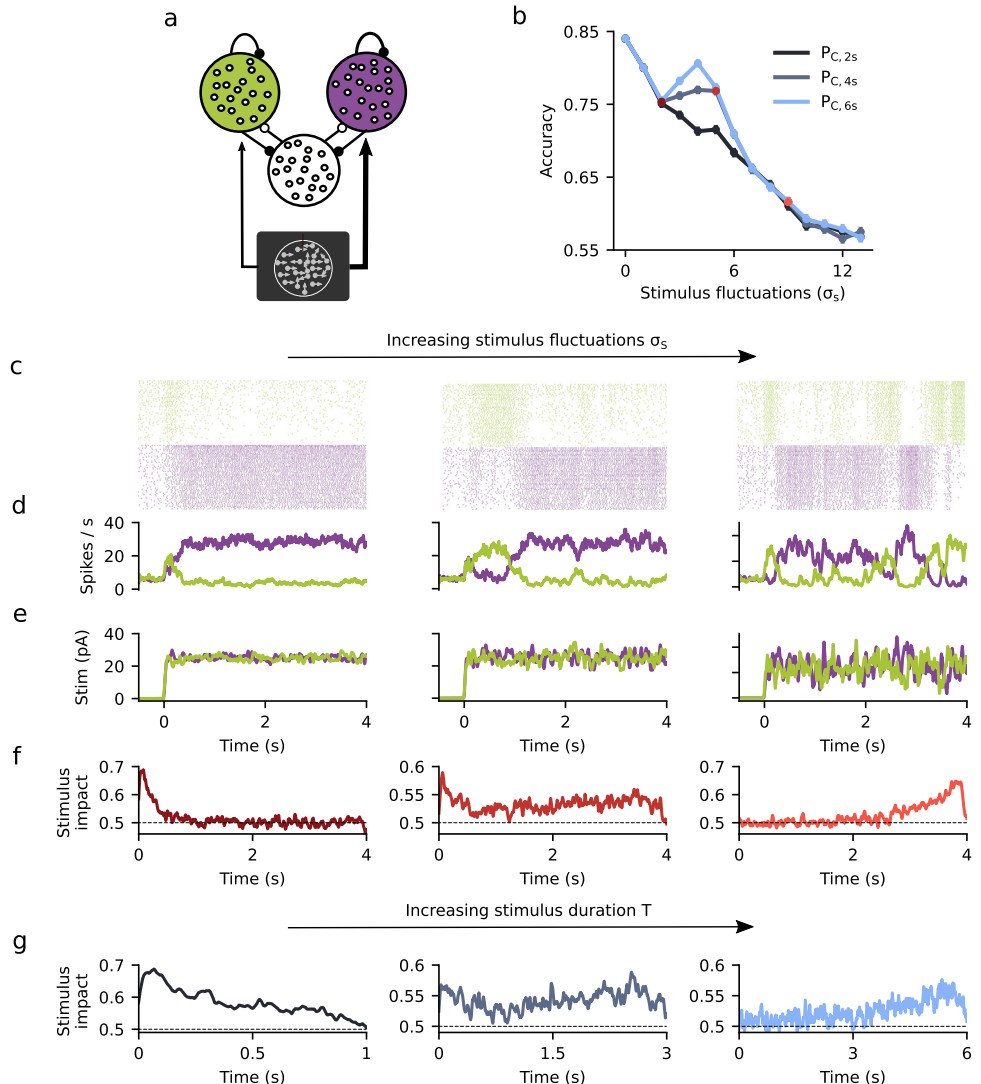

**Fig. 5 Signatures of flexible categorization dynamics in a spiking network. a** Schematic of the spiking network consisting of two stimulus-selective populations (green and purple) made of excitatory neurons that compete through an untuned inhibitory population (white population). **b** Accuracy $P_C$ versus stimulus fluctuations $\sigma_S$ obtained from simulations of the spiking network for three values of the stimulus duration $T = 2$, 4, and 6 s (see inset). **c–e** Single-trial examples showing spike rastergram from the two excitatory populations (1000 + 1000 neurons) (**c**), traces of the instantaneous population rates (count window 30 ms) (**d**) and of the input stimuli (**e**), for different values of stimulus fluctuations $\sigma_S = 2$ (left), 4.5 (middle), and 9 pA (right). Colored points in (**b**) indicate the $\sigma_S$ used. **f** Psychophysical kernels obtained for each $\sigma_S$ value. The mean stimulus input was $\mu = 0.015$ and the stimulus duration $T = 4$ s. **g** Psychophysical kernels for $\sigma_S = 5$ pA and different stimulus duration $T = 1$, 3, and 5 s, from left to right.

monotonicity in the psychometric curve, it was the difference between two transition probabilities, the transition probability with internal noise versus the probability without internal noise, that was maximal when consistency decreased (Fig. 4b). Also as before, to observe the non-monotonicity in the consistency, $\sigma_I$ had to be sufficiently small not to cause transitions on its own (Fig. 4a, b). Notice however that the non-monotonicity here was not caused by the asymmetry between correcting versus error transitions, as consistency was computed using $\mu = 0$ stimuli (i.e., there was no correct choice). The effect was a result of the nonlinear attractor dynamics of the DWM and thus it could not occur in any of the canonical models (Fig. 4c).

**Flexible categorization in a spiking network with attractor dynamics**. Having shown that the DWM generates signatures of attractor dynamics which are qualitatively different from any canonical model, we then assessed whether these could be

reproduced in a more biophysically realistic network model composed of leaky integrate-and-fire neurons (Methods). The network consisted of two populations of excitatory (E) neurons ($N_E = 1000$ for each population), each of them selective to the evidence supporting one of the two possible choices, and a nonselective inhibitory population ($N_I = 500$) (Fig. 5a). The network had sparse, random connectivity within each population (probability of connection between neurons was 0.1) and neurons were coupled through current-based synapses with exponential decay. The stimulus was modeled as two fluctuating currents, reflecting evidence for each of the two choice options and injected into the corresponding E population. The two currents were parametrized by their mean difference $\mu$ and their standard deviation $\sigma_S$ (see Methods). In addition, all neurons in the network received independent stochastic synaptic inputs from an external population. As in previous attractor network models used for stimulus categorization, the two E populations competed through the inhibitory population[19]. Thus, upon presentation of

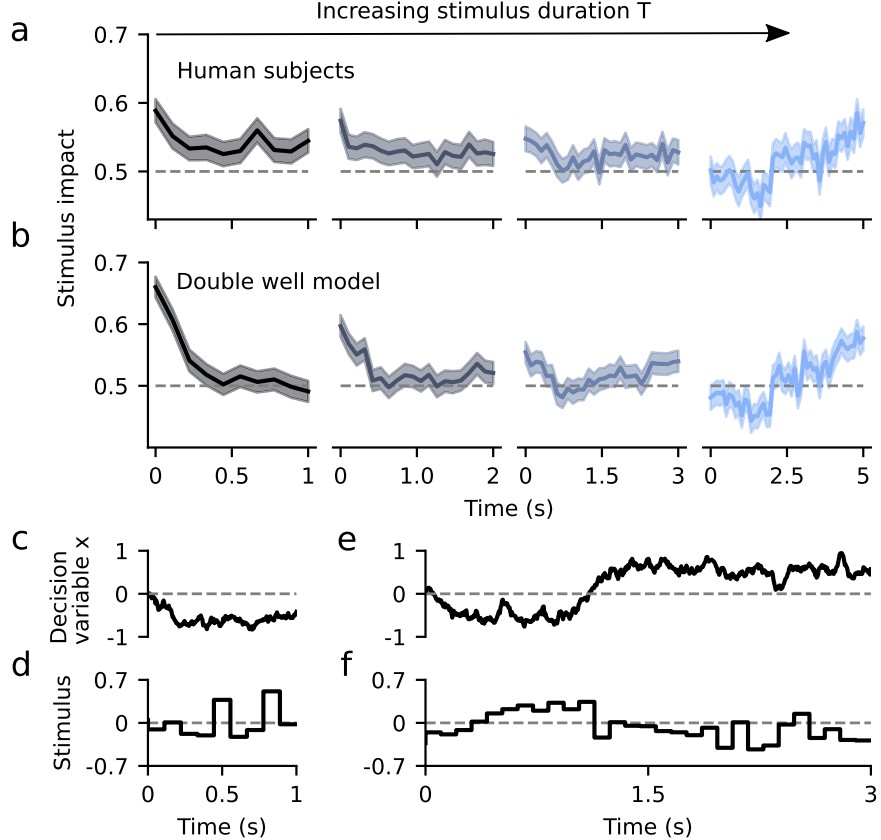

**Fig. 6 The double well model accounts for experimentally observed changes in psychophysical kernels. a** Psychophysical kernels for different stimulus durations, obtained from human subjects performing a brightness discrimination task ($N = 21$)[34]. From left to right, stimulus duration was $T = 1, 2, 3$, and 5 s. **b** Psychophysical kernels obtained by fitting the DWM to categorize the very same stimuli presented to the human subjects (i.e., same temporal fluctuations of net evidence; see Methods). Lines represent the kernels obtained from pooling all data across subjects and the error bands represent s.e.m. **c–f** Example traces of the decision variable of the fitted DWM (**c**, **e**) and the stimulus (**d**, **f**) for 1 and 3 s trials. Notice that the stimulus fluctuations mimicked the visual stimulus which was made of time frames of 100 ms.

an external stimulus, there were two stable solutions: one solution in which one E population fired at a high rate while the other fired at a low rate and vice versa (Fig. 5). Notice that in contrast with the DWM in which the noise was white (i.e., temporally uncorrelated), in this network the external noise was colored (stimulus was an Ornstein–Uhlenbeck process with $\tau_{stim} = 20$ ms) and the internal fluctuations reflected the stochasticity of the spiking network dynamics which are strongly affected by the synaptic time scales. Similar to the DWM, we found a non-monotonic relation between the accuracy and the magnitude of the stimulus fluctuations $\sigma_S$ provided the stimulus duration $T$ was sufficiently long (Fig. 5b). Moreover, as $\sigma_S$ increased the integration regimes of the network changed from primacy to recency, passing through the flexible categorization regime (Fig. 5c–f). In this regime, transitions between attractor states occurred when there were input fluctuations that extended over hundreds of milliseconds, indicating that the temporal integration of evidence continued even after one of the attractors was reached (Supplementary Fig. 2b). The crossover between primacy and recency regimes was also observed at constant $\sigma_S$ when we varied the stimulus duration $T$ (Fig. 5g). We went one step further in including biophysical detail and confirmed that a conductance-based spiking neural network model with explicit AMPA, GABA, and NMDA receptor dynamics[19] showed qualitatively the same behavior (Supplementary Fig. 4). Thus, the signatures of attractor dynamics that we had identified did not depend on the simplifying assumptions of the DWM and could be replicated in an attractor network with more biophysically plausible parameters.

**Changes in PK with stimulus duration in human subjects unveiled the flexible categorization regime.** We tested whether the DWM could parsimoniously account for the variations of the integration dynamics previously found in a perceptual categorization task as the stimulus duration was varied[34]. In the experiment, human subjects had to discriminate the brightness of visual stimuli of variable duration $T = 1, 2, 3$, or 5 s. Confirming previous analyzes[34], the average PKs across subjects changed from primacy to recency with increasing stimulus durations (Fig. 6a). To assess whether these changes in the shape of the PKs could be captured by the DWM, we used the DWM to categorize the same stimuli (the exact same temporal stimulus fluctuations and number of trials; see Methods) that were presented to the human subjects (Fig. 6c–f). We found that the PKs for different stimulus durations obtained in the DWM were very similar to the experimental data (Fig. 6b). Importantly, these results were obtained with fixed model parameters for all stimulus durations suggesting that the variation in PK did not necessarily indicate a change of the integration mechanism of the model, as previously suggested[34]. Rather, fixed, but nonlinear attractor dynamics in the DWM parsimoniously accounted for the observed PK changes.

**Stimulus integration across a memory period is consistent with flexible categorization dynamics.** Finally, we tested the DWM in a task that requires evidence accumulation and working memory. We used published data from two studies carrying out a psychophysical experiment in which subjects had to categorize the

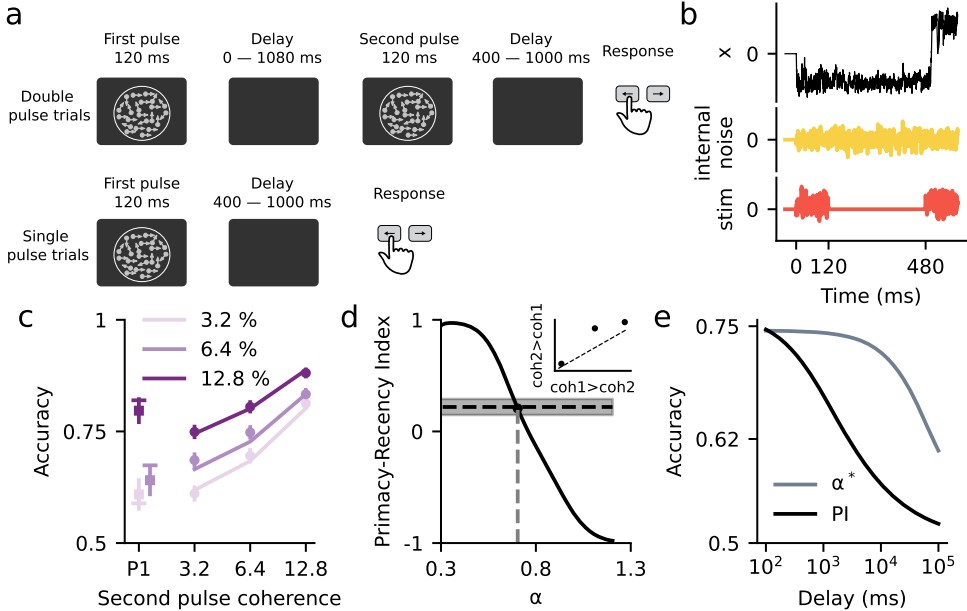

**Fig. 7 The flexible categorization regime accounts for the combination of two pulses of evidence during a working memory task. a** Visual motion categorization experiment consisting of interleaved double pulse (top) and single pulse trials (bottom)[35,40]. On double pulse trials, the two motion pulses were separated by a variable delay (duration 0, 120, 360, or 1080 ms). Coherences were randomly selected from trial-to-trial. In two pulse trials, they could be different but the motion direction was always congruent ($N = 9$). **b** Traces of the decision variable of the DWM (black), the internal fluctuations (yellow), and the stimulus (orange) for an example double pulse trial. **c** Accuracy for single (squares) and two pulse trials (dots) versus the coherence of the second pulse observed in the data from[35,40] (dots) and the values obtained from the fitted DWM (lines). Because accuracy in the experiment did not depend on delay length, dots show the average accuracy across all delays. Different colors represent different first pulse coherences (see inset). Symbols show mean across subjects and error bars show 95% confidence intervals. **d** Primacy-recency index (PRI) for the DWM as a function of the barrier height ($c_2$). The black dot marks the PRI for the fitted parameter $\alpha^\star = 0.7$. The horizontal line is the PRI computed from the psychophysical data (gray area 95% confidence interval). Inset: accuracy for two pulse stimuli in which coherence is larger in pulse 2 than in pulse 1 (i.e., coh2 > coh1) versus accuracy for the same pulses presented in the reverse order (i.e., coh1 > coh2). Consistent with the recency effect, accuracy is slightly better for coh2 > coh1 stimuli. **e** Accuracy as a function of the delay duration for DWM and for the Perfect Integrator. In the DWM, which used the fitted parameter $\alpha^\star$ and $\sigma_I = 0.32$ and $\sigma_S = 0.40$, the accuracy is independent of the delay up to 1 s. In contrast, for the same internal noise $\sigma_I$, the accuracy of the perfect integrator decreases continuously for all delays.

motion direction of a random dot kinematogram[35,40]. Interleaved with the trials showing a single kinematogram (single pulse trials, duration 120 ms) there were also trials having two kinematograms separated by a temporal delay (two pulse trials). In these two pulse trials, subjects had to combine information from both pulses in order to categorize the average motion direction. The two pulses could have different motion coherence but they always had the same motion direction (Fig. 7a). Subjects were able to combine the evidence from the two pulses and their accuracy did not depend on the duration of the delay period for durations up to 1 s, meaning that they were able to maintain the evidence from the first pulse without memory loss. Overall, subjects gave slightly more weight to the second than the first pulse (Primacy-Recency Index = 0.22; see Methods). Qualitatively, the DWM could in principle capture this behavior because its underlying dynamics can solve the two parts of the task, the maintenance of information during the working memory period and the combination of the two pulses of evidence (Fig. 7b). The model would categorize the first pulse in one of the attractors, which would be stably maintained during the delay because the internal noise is insufficient to cause transitions. Finally, given the asymmetry in the DWM transition rates (Fig. 3c), the second pulse could reverse incorrect initial categorizations while minimizing the risk of erroneously reversing correct ones (Fig. 7b). To assess whether the DWM could indeed fit the data quantitatively, we computed the accuracy for each stimulus condition using Kramers' transition rate theory and fitted the parameters using maximum likelihood estimation (solid lines, Fig. 7c; Methods). We found that

the DWM could fit the accuracy across conditions quite accurately (Fig. 7c). Interestingly, the fitted DWM worked close to the flexible categorization regime, matching the slight recency effect coming out from the combination of the two pulses (Fig. 7d).

Because subjects' accuracy did not depend on delay duration[35,40], the model fitting could only determine the value of the sum of the stimulus and internal noises $\sigma = \sqrt{\sigma_I^2 + \sigma_S^2}$ and set an upper bound $\sigma_I^{max}$ for the internal noise: for any value $\sigma_I \le \sigma_I^{max}$ the transitions during the delay were negligible (<1%) and the DWM yielded the same behavior (see Methods). Choosing the $\sigma_I$ to be at the upper bound $\sigma_I^{max}$, yielded a constant accuracy for delays up to ~1 s. For longer delays, however, transitions during the delay became active causing forgetting and accuracy decrease (Fig. 7e) as has been shown in experiments using a broader range of delays[41]. In contrast, the perfector integrator did not show a range of delays over which the accuracy remained constant (Fig. 7e): the internal noise had a much larger impact on the maintenance of stimulus evidence so that, for any significant level of internal noise, the accuracy decreased continuously with delay duration. In total, our analysis shows that the DWM can quantitatively fit psychophysical data from a working memory task, and that longer delays could provide a qualitative test for the model.

## Discussion
We have investigated the attractor model with winner-take-all nonlinear dynamics and we have found new, experimentally

testable signatures that can distinguish it from the other models. First, the attractor model exhibits a continuous crossover from the primacy regime[19,23] to the recency regime. Between these two regimes we found the new flexible categorization regime in which the integration of stimulus fluctuations was maximally extended over time (Fig. 2 and Supplementary Fig. 2). Second, in this regime a qualitative asymmetry between correcting and error transitions gave rise to a non-monotonic psychometric curve (Fig. 3). Third, the rapid activation of transitions between decision states with the stimulus fluctuations also caused an unexpected non-monotonic dependence of the stimulus consistency (Fig. 4a). Finally, we used two previous psychophysical experiments to show that the attractor model can quantitatively fit variations in PK profile with stimulus duration (Fig. 6) and fit categorization accuracy in a task with integration of evidence across memory periods (Fig. 7).

Recently, two studies have proposed alternative models that can explain the differences of PK time-courses found across subjects and experiments. In the first model, based on approximate Bayesian inference, the primacy effect produced by bottom-up versus top-down hierarchical dynamics, was modulated by the stimulus properties which could yield different PK time-courses, a prediction that was tested in a visual discrimination task[42]. The second study proposed a model that can produce different PK time-courses by adjusting the time scales of a divisive normalization mechanism, which yields primacy, and a leak mechanism, which promotes recency[43]. In addition, this model can also account for bump shaped PKs, a class of PK that was found together with primacy, recency, and flat PKs, in a study carried out using a large cohort of subjects (>100)[44]. In the attractor model, the differences in the PK found across subjects or fixed stimulus properties could be explained by individual differences in the shape of the potential. Specifically, differences in the height of the barrier between the two attractor states would generate a variety of PK time-courses (Fig. 7c) as the integration regime ultimately depends on the ratio between the total noise $\left(\sigma_S^2 + \sigma_I^2\right)$ and the height of the barrier. A natural extension of our approach would be to assume that a time-varying process during the trial, e.g., an urgency signal[45], can progressively modify the shape of the potential. In that case, the DWM with an urgency signal that changed the shape of the potential from a single well at stimulus onset into a double well at stimulus offset could readily reproduce the bump shaped PKs (not shown) recently reported[44]. In sum, the attractor model shows a large versatility generating the diversity of PK shapes reported in the literature[8,10,12,15–17,44]. Although several distinct models can account for the variety of PK shapes, they rely on a variety of neural mechanisms. Future electrophysiological or psychophysical experiments where the different models predict qualitatively different results will help distinguish between these possible mechanisms.

It has been previously shown that noise, from the stimulus or internal sources, can increase the accuracy of an attractor model with three stable attractors (i.e., with multistability): an undecided state and two decision states[46,47]. In this model, the decision variable starts in the undecided state and, if it does not escape from this state during the stimulus presentation, the decision is made randomly. Thus, the noise can allow the decision variable to escape from the undecided state and increase the accuracy. Here, we have studied the attractor model in the winner-take-all regime, i.e., without an undecided state, and we have found that it is the large difference between the rate of correcting and error-generating transitions that produces the increase in accuracy in the flexible categorization regime. This is conceptually very different from transitions between the undecided state to the decision states. The same mechanism presented here drives the classic stochastic resonance[48] where a particle moving in a double well

potential driven by a periodic signal necessitates of a suitable magnitude of noise for the system to follow the signal (i.e., escape from the well when it is no longer the global minimum). Similar to the effect described with the multistable attractor model[46], the accuracy decreases to chance in the deterministic noiseless case ($\sigma = 0$). In contrast, the accuracy for the DWM is greatest for $\sigma = 0$ because the initial position of the decision variable ($x_0 = 0$) belongs to the basin of attraction of the correct attractor and thus it always rolls down to the correct attractor. However, whether this bump in accuracy produced by the attractor model as a function of the stimulus fluctuations ($\sigma_S$) is a local or a global maximum, or if it exists at all, depends on internal parameters such as the internal noise ($\sigma_I$) or the height of the barrier. These internal parameters can be different for different subjects and thus, one should expect to find this non-monotonic psychometric curve only in a fraction of subjects. Indeed, we carried out a visuospatial binary categorization task in which the fluctuations of the evidence $\sigma_S$ were varied systematically from trial-to-trial. Preliminary analysis shows that the majority of subjects display a psychometric curve $P(\sigma_S)$ with a plateau followed by a decay as $\sigma_S$ increased. A fraction of subjects exhibited however a non-monotonic dependence but the dependence of PK and other aspects of their behavior (e.g., idiosyncratic biases) on $\sigma_S$ were not fully captured by the DWM dynamics. A future study will extend the DWM so that it can capture these data.

The key mechanism underlying the flexible categorization regime are the transitions between attractor states which, functionally, can be viewed as changes of mind[5,49]. Changes of mind have been previously inferred from sudden switches in the direction of the motor response[5,49] but also from decision bound crossings of the decision variable read out from neuronal population recordings[50–53]. In reaction time tasks, an extension of the drift diffusion model can fit the modulation of the probability of observing a change of mind as a function of the mean stimulus strength[5]. In this model, a first crossing of the decision bound initializes the response that is reversed if the decision variable crosses the opposite bound before the motor response is completed. As in the DWM, this model predicts that correcting changes of mind are more likely than error changes of mind. However, this asymmetry does not imply a non-monotonic accuracy with the stimulus fluctuations in a fixed duration task. This is because in the linear DDM with changes of mind[5], the correcting transition probability $p_C$ is not exponentially more likely than error transitions as in the DWM (Eq. 20). Thus, the benefit of having more correcting transitions as $\sigma_S$ increases does not offset the cost of decreasing the signal-to-noise ratio (not shown). An attractor network has also been used previously to explain changes of mind during the motor response[54]. Our work extends this study in several ways, by characterizing the full spectrum of integration regimes in the attractor model and by showing qualitative experimentally testable signatures of decision state transitions (e.g., non-monotonicity in the accuracy and coherence versus $\sigma_S$). One interesting question is whether correcting changes of mind could generate similar nonlinear effects as those reported here (Fig. 3a, b) in tasks with n > 2 choices. A preliminary analysis using rate-based networks suggests that this is in fact the case (Supplementary Fig. 5). We simulated rate networks composed of $n$ excitatory populations competing with each other via mutual inhibition and found that in the winner-take-all regime, strong stimulus fluctuations causing attractor transitions could have a beneficial effect and yield a non-monotonic psychometric curve $P(\sigma_S)$ (Supplementary Fig. 5c). Thus, although a more detailed analysis of these multiple-choice networks is needed, these examples suggest that the asymmetry between correcting and error transitions underlying the raise in accuracy with $\sigma_S$, was a general mechanism that may be in play in tasks with more than two choices.

An important question in perceptual decision making is the extent to which subjects can integrate evidence during the stimulus presentation. It has been recently pointed out that differentiating between integrating and non integrating strategies may be more difficult than naively thought[55]. Here we evaluate the degree of evidence integration using the PK area. In the flexible categorization regime this area is maximum, and the DWM can integrate a large fraction of stimulus fluctuations (Fig. 2f). Indeed, we have shown that in this regime, the spiking network model, built of neural units with time-constants of 20 ms, could generate transitions by integrating fluctuations over hundreds of milliseconds (Supplementary Fig. 2b). Further work would be required to quantitatively characterize the emergence of this slow integration time-scale. The PK area however, is not a measure of accuracy, when accuracy is defined as the ability to discriminate the sign of the mean stimulus evidence, $\mu$. Thus, the accuracy in the DWM is maximal for $\sigma_S \approx 0$ (Fig. 3b) but the area is close to zero (Fig. 2f). This mismatch simply reflects that, in the absence of internal noise, the task does not require integration of the stimulus fluctuations. However, if we only considered stimuli with $\mu = 0$ and we defined the stimulus category based on the sign of stimulus integral, the accuracy would be strongly correlated with the PK area and it would be maximal in the flexible categorization regime.

Finally, equipped with the theoretical results on the attractor model, we have revisited two psychophysical studies seeking for signatures of attractor dynamics. With the data from the first study[34], we have tested a key prediction of the attractor models and have shown that the DWM can readily fit the crossover from primacy, to flexible categorization, to recency observed in the data as stimulus duration increases. This fit shows that the behavioral data in this task is consistent with the presence of transitions between attractor states during the perceptual categorization process (Fig. 6). We used psychophysical data from two other studies[35,40], to show that in a regime close to the flexible categorization the DWM could fit the categorization accuracy as a function of stimulus strength for all memory periods (Fig. 7). Thus, the described asymmetry between correcting and error transitions allowed the DWM to combine evidence from the two pulses and yield a higher accuracy than a single pulse, just like subjects did (Fig. 7b, compare single vs two pulse trials using the same coherence, e.g., 6.4% versus $6.4 + 6.4$%). Models that assume perfect integration of evidence can generally store a parametric value in short-term memory but they are susceptible to undergoing diffusion over time, causing a drop in memory precision as the delay increases[56,57]. In contrast, the fact that the accuracy did not decrease with delay duration suggests that the information stored in memory could be categorical instead of parametric[58,59], a feature naturally captured by the DWM (Fig. 7d). Alternatively, it could reflect a parametric memory with negligible internal noise[60]. Interpreting neural recordings can also be non-conclusive as different areas can simultaneously represent stimulus information with different levels of categorization[61]. To overcome these shortcomings in understanding whether the stored information is categorical or parametric, we propose an experiment that combining electrophysiology with psychophysics can qualitatively distinguish between these two alternatives (see Supplementary Fig. 6). An alternative version of the DDMA model where the sensitivity to the second pulse was larger than to the first one could also account for the combination of the two pulses[40]. This feature captured the slight recency effect found in the data, but it left unanswered the key question of why the subjects did not use their maximum sensitivity during the first pulse. In total, our findings provide evidence that an attractor model, working in the flexible categorization regime, can capture aspects of the data that were previously viewed as incompatible

with its dynamics, and propose a series of testable predictions that may further shed light onto the brain dynamics during sensory evidence integration.

## Methods

**Model simulations**. For all simulations, we solve the diffusion Eq. 2 using the Euler method:

$$x(t+1) = x(t) - \frac{\Delta t}{\tau} d\varphi(x(t))/dx + \sqrt{\frac{\Delta t}{\tau}}(\sigma_I \xi_I(t) + \sigma_S \xi_S(t)), \qquad (6)$$

with $\Delta t = \tau/40$. The time constant $\tau$ of the DWM was chosen to be 200 ms to represent the effective integration time constant that emerges from the dynamics of a network[19].

We summarized the parameters used in each figure in Table 1 (Supplementary Information).

In Fig. 4, we use stimuli with exactly zero integrated evidence, $\int S(t)dt = 0$. For each stimulus i, we first created a stream of normal random variables $y_i(t)$. Then we z-score $y$ and we multiplied by $\sigma_S$:

$$S_i(t) = \sigma_S \frac{y_i(t) - \hat{y}}{\sigma_y}. \qquad (7)$$

After this transformation, the mean and standard deviation of $S_i$ are exactly 0 and $\sigma_S$ respectively.

**Psychophysical kernel**. We measure the impact of stimulus fluctuations during the course of the trial on the eventual decision by means of the so-called PK. Put simply, given a fixed mean signal, some stimulus realizations may favor a rightward choice (say a positive decision variable) and others a leftward one. If this is the case, and we sort the stimuli over many trials by decision, we will see a clear separation which can be quantified via a ROC analysis. Mathematically, for each trial i, we subtract the mean evidence ($\mu_i$) of each trial $s_i(t) = \mu_i + \sigma_S \xi_i$ to avoid that the distributions of stimuli that produce left and right choices are trivially separated by their mean evidence:

$$\hat{s}_i(t) = s_i(t) - \mu_i. \qquad (8)$$

Thus $\hat{s}_i(t) = \sigma_s \xi_i$ are simply the stimulus fluctuations. Then, for each time $t$, we compute the probability distribution function of the stimuli that produce a right ($f(\hat{s}_R(t))$) or left ($f(\hat{s}_L(t))$) choice. The PK is the temporal evolution of the area under the ROC curve between these two distributions

$$PK(t) = auc(f(\hat{s}_R(t)), f(\hat{s}_L(t))). \qquad (9)$$

**Normalized PK area and slope**. In order to quantify the magnitude and the shape of a PK, we defined two measures, the PK area and the PK slope:

1) The normalized PK area is a measure of the overall impact of stimulus fluctuations on the upcoming decision, it ranges from 0 (no impact) to 1 (the stimulus fluctuations are perfectly integrated to make a choice). It is defined as

$$NPKA = \frac{\int_0^T PK(t) - 0.5 \, dt}{\int_0^T PK_{PI}(t, \sigma_i = 0) - 0.5 \, dt}, \qquad (10)$$

where $T$ is the stimulus duration. $NPKA$ is the PK area normalized by the PK area of a PI in the absence of internal noise ($\sigma_i = 0$), i.e., an ideal observer.

2) The normalized PK slope is the slope of a linear regression of the PK, normalized between $-1$ (decaying PK, primacy) to $+1$ (increasing PK, recency). Because we wanted the PK slope to quantify the shape of the PK rather than its magnitude (which is captured by the PK area), we first normalized the PK to have unit area,

$$NPK(t) = \frac{PK(t) - 0.5}{\int_0^T PK(t) - 0.5 \, dt}, \qquad (11)$$

where $T$ is the stimulus duration. We then fit the NPK with a linear function of time,

$$LPK(t) = \beta_0 + k\beta_1 \times t, \qquad (12)$$

where $\beta_1$ is the PK slope and $k = \frac{1}{2 \cdot var(t)}$ is a factor that normalizes the PK slope to the interval $(-1, +1)$.

**Accuracy for the DWM**. To compute the accuracy for the DWM, we assume that the time spent in the unstable region is much shorter than the time spent in one of the attractors. This assumption allows us to treat the system as a Continuous Markov Chain (CMC) with only two possible states correct and error. The first step is to compute the probability of first visiting the correct attractor which will be used as the initial state of the CMC[62]

$$P_0 = \frac{\int_{x_E}^{x_0} exp\left(\frac{2\varphi(x)}{\sigma_I^2 + \sigma_S^2}\right) dx}{\int_{x_E}^{x_C} exp\left(\frac{2\varphi(x)}{\sigma_I^2 + \sigma_S^2}\right) dx}, \qquad (13)$$

where $\varphi$ is the potential in Eq. 3, $x_C$ and $x_E$ are the $x$ values of the correct and error attractors whereas $x_0 = 0$ is the initial position of $x$. The integrals of $P_0$ can be computed assuming that the term $x^4$ is very small for values of $x_0 \simeq 0$:

$$P_0 = \frac{erf\left(\frac{\sqrt{2\alpha}}{\sigma}\left(x_0 + \frac{\mu}{2\alpha}\right)\right) - erf\left(\frac{\sqrt{2\alpha}}{\sigma}\left(x_E + \frac{\mu}{2\alpha}\right)\right)}{erf\left(\frac{\sqrt{2\alpha}}{\sigma}\left(x_E + \frac{\mu}{2\alpha}\right)\right) - erf\left(\frac{\sqrt{2\alpha}}{\sigma}\left(x_C + \frac{\mu}{2\alpha}\right)\right)}. \quad (14)$$

The second step is to compute the correcting and error transition rates[37,62]

$$k_C = \frac{\sqrt{|\varphi''(x_E)\varphi''(x_U)|}}{2\pi} exp\left(-\frac{2(\varphi(x_U) - \varphi(x_E))}{\sigma_I^2 + \sigma_S^2}\right) \quad \text{and} \quad (15)$$

$$k_E = \frac{\sqrt{|\varphi''(x_C)\varphi''(x_U)|}}{2\pi} exp\left(-\frac{2(\varphi(x_U) - \varphi(x_C))}{\sigma_I^2 + \sigma_S^2}\right), \quad (16)$$

where $x_U$ is the $x$ position at the unstable state. These are the transition rates of a Continuous Markov Chain with only two states: correct and incorrect. The probability of making a correcting and error-generating transition during a trial are[63]:

$$p_C(T) = P_\infty(1 - exp(-kT)), \quad (17)$$

$$p_E(T) = (1 - P_\infty)(1 - exp(-kT)), \quad (18)$$

where $k = k_C + k_E$, $T$ is the stimulus duration and $P_\infty = \frac{k_C}{k_C + k_E}$ is the probability of the stationary state being the correct one ($T \to \infty$). Finally, the probability of being in the correct attractor given the model and stimulus parameters is

$$P = P_0(1 - p_E) + (1 - P_0)p_C. \quad (19)$$

The probability of correct is the probability to first visit the correct attractor and remain in it ($P_0(1-p_E)$) plus the probability to first visit the error attractor and correct the initial decision ($(1-P_0)p_C$). To be more quantitative, we can compute the ratio between the probability of a correcting (Eq. 17) and an error-generating transition (Eq. 18):

$$p_C/p_E \propto exp\left(2(\varphi(x_E) - \varphi(x_C))/\sigma^2\right) \quad (20)$$

For small values of the mean signal $\mu \ll 1$, we can rewrite the ratio between the correcting and error-generating transitions as a function of the potential parameters. To this aim we compute the fixed points of order $\vartheta(\varepsilon^2)$ using $\mu = \varepsilon\overline{\mu}$ where $\overline{\mu}$ is a parameter of order 1 and $x = x_0 + \varepsilon x_1$:

$$x_C = \sqrt{\frac{\alpha}{2}} + \frac{\mu}{4\alpha} + \vartheta(\varepsilon^2), \quad (21)$$

$$x_E = -\sqrt{\frac{\alpha}{2}} + \frac{\mu}{4\alpha} + \vartheta(\varepsilon^2) \quad \text{and} \quad (22)$$

$$x_U = -\frac{\mu}{4\alpha} + \theta(\varepsilon^2), \quad (23)$$

where $x_U$ is the $x$ position of the unstable state (note that $x_U = 0$ when $\mu = 0$) and $x_C(x_E)$ is the position of the correct (error) attractor. Using these fixed points, the ratio between the correcting and error-generating transitions is

$$p_C/p_E = exp\left(\frac{4\mu}{\sigma^2}\sqrt{\frac{\alpha}{2}}\right). \quad (24)$$

Which shows that the ratio between correcting transitions and error-generating ones increases exponentially with the mean stimulus ($\mu$) as long as stimulus fluctuations are not too large. These probabilities are illustrated in Fig. 3c, $p_C$ increases steeply as a function of stimulus fluctuations even before $p_E$ reaches non-negligible values and for large stimulus fluctuations both probabilities tend to 0.5.

To find the maximum of the accuracy, we derive Eq. 19 respect to $\sigma$:

$$\frac{dP}{d\sigma} = -\frac{2}{\sigma^3}\frac{d}{d\beta}[P_0 exp(-kT) + p_C], \quad (25)$$

where we rewrite Eq. 19 as a function of $p_C$ and $P_0$. From this equation, it can be shown (see Supplementary Information) that the local maximum in accuracy is

$$\sigma_{IC}^2 = \frac{\alpha}{2}\frac{1}{log\left(\frac{2T\alpha}{\pi z_0}\right)}, \quad (26)$$

and the critical value of $\mu$ above which the accuracy decreases monotonically with the stimulus fluctuations is

$$\mu_C = \frac{\alpha}{2}\sqrt{\frac{\alpha}{2}}. \quad (27)$$

## Spiking network

*Network model.* In Fig. 5 we consider a network of randomly connected current-based integrate-and-fire neurons, similar to[28]. The conductance-based all-to-all connected network shown in Supplementary Fig. 4 was exactly the original network model presented in ref. [19]. The current-based network consists of two populations of excitatory neurons (A and B), both of which are recurrently coupled between them and to a population of inhibitory interneurons (I). We study the case in which the system is near a steady bifurcation to a winner-take-all state. It is in the vicinity of the bifurcation that the dynamics of the network can be captured in a one-dimensional amplitude equation which describes the slow evolution along the critical manifold[28]. The evolution of the membrane potential $V_i^X(t)$ from the i-th neuron in population $X$ is given by:

$$\tau_m^E \frac{dV_i^A}{dt} = -(V_i^A - E_l) + I_i^{AA} - I_i^{AI} + I_i^{Aext}/g_L, \quad (28)$$

$$\tau_m^E \frac{dV_i^B}{dt} = -(V_i^B - E_l) + I_i^{BB} - I_i^{BI} + I_i^{Bext}/g_L, \quad (29)$$

$$\tau_m^I \frac{dV_i^I}{dt} = -(V_i^I - E_l) + I_i^{IA} + I_i^{IB} + I_i^{Iext}/g_L, \quad (30)$$

where the synaptic input voltages of the form $I_{XY}$ indicate interactions from neurons in population $Y$ to neurons in population $X$, while external synaptic inputs are given by $I^{Xext}$. The synaptic inputs are sums over all postsynaptic potentials (PSPs), modeled as exponential functions with a delay. The synaptic inputs take the form

$$I_i^{XY} = \sum_j J_{ij}^{XY} g_{ij}^{XY}. \quad (31)$$

The dynamics of excitatory and inhibitory synapses are described by

$$\tau_s^Y \frac{dg_{ij}^{XY}}{dt} = -g_{ij}^{XY}, \quad (31)$$

After the presynaptic neuron $j$ fires a spike at time $t_k^{XY}$, the corresponding dynamic variable is incremented by one at $t_k^{XY} + \delta_k^Y$, that is after a delay $\delta_k^Y$.

External synapses have instantaneous dynamics

$$I_i^{Iext} = \sum_j J_{ij}^{ext} \sum_k \delta\left(t - t_{k,j}^{Xext}\right), \quad (33)$$

i.e., a presynaptic action potential from neuron $j$ of the external population at time $t_{k,j}^{Xext}$ results in an instantaneous jump of the external synaptic input variable. A spike is emitted whenever the voltage of a cell from an excitatory (inhibitory) population crosses a value $\Theta$, after which it is reset to a reset potential $E_r$.

We consider the case of sparse random connectivity for which, on average, each neuron from population X receives a total of $C_{XY}$ synapses from population Y. The pairwise probability of connection is thus $\epsilon_{XY} = C_{XY}/N_Y$, where $N_A = N_B = N_E$ and $N_I$ are the number of neurons in the respective populations. For nonzero synapses we choose $J_{ij}^{AA} = J_{ij}^{BB} = J_{EE}$, $J_{ij}^{IA} = J_{ij}^{IB} = J_{IE}$ and $J_{ij}^{AI} = J_{ij}^{BI} = J_{EI}$.

The stimulus input current is modeled similar to[23], with the exact same stimulus input being injected to each neuron in each of the two excitatory populations. The stimulus input onto each of the excitatory populations A and B is given by

$$I_{stim}^A(t) = I_0(1 + \mu) + \sigma_S z^A(t), \text{ and} \quad (34)$$

$$I_{stim}^B(t) = I_0(1 - \mu) + \sigma_S z^B(t), \quad (35)$$

where the first term describes the mean stimulus input onto each population and the second term the temporal modulations of the stimulus with standard deviation $\sigma_{stim}$. The term $\mu$ parametrizes the mean difference of the two stimulus inputs and it captures the amount of net stimulus evidence favoring one choice over the other (i.e., $\mu = 0$ represents an ambiguous stimulus with zero mean sensory evidence). Finally, $z^A(t)$ and $z^B(t)$ are independent realizations of an Ornstein–Uhlenbeck process, defined by $\tau_{stim}\frac{dz}{dt} = -z + \sqrt{2\tau_{stim}}\xi(t)$, where $\xi(t)$ is Gaussian white noise (mean 0, variance dt).

*Simulation details.* The network model was implemented in Python 3 using the Brian 2 simulator version 2.3[64]. We used the Euler integration method with a time step of 0.1 ms. We simulated fixed duration trials of varying stimulus duration. Stimulus presentation was preceded by a 500 ms interval to prevent transient effects due to initial conditions. The choice outcome of the network was determined by the neural population with a higher population firing rate over the last 100 ms of the stimulus period. Results for a given stimulus condition ($\sigma_S$ and $T$) are based on 5000 trials using different realizations of the network connectivity, random initial conditions as well as different realizations of the external background inputs into each circuit. The value of all the parameters can be found in Table 2 (Supplementary Information)

**Psychophysical data and model fitting.** In Fig. 6, we used data from experiments 1 and 4 from[34] with a total of $N = 21$ humans subjects ($N = 13$ in experiment 1 and $N = 8$ in experiment 4). The data can be accessed here: https://doi.org/10.1371/journal.pcbi.1004667. The stimuli consisted of two brightness-fluctuating round

disks. In each stimulus frame (duration 100 ms), the brightness level of each disk was updated from one of two generative Gaussian distributions that had the same variance but different mean: either one distribution had a high mean value and the second a low value or vice versa. At the end of the stimulus, the subjects had to report the disk with a higher overall brightness (i.e., which disc corresponded to the generative distribution with higher mean). Incorrect responses were followed by an auditory feedback. Trials were separated into five equal length segments, in 80% of the trials, a congruent or incongruent pulse of evidence was presented at a random segment. This increase or decrease of evidence was corrected in the rest of the segments and as a consequence the stimuli were anticorrelated. In experiment 1 stimuli with 1, 2, or 3 s duration were presented in blocks of 60 trials whereas in experiment 4, the stimulus duration was 5 s. We computed the PK using the procedure described above (see section Psychophysical kernel) but first computing the difference in brightness of the two disks. We also subtracted the mean difference in order to have a one-dimensional stimulus trace with zero mean. Namely

$$S^i(t) = S_R^i(t) - S_L^i(t) - (\mu_R^i - \mu_L^i), \tag{36}$$

where $S_L^i(t)$ is the brightness of the $t$-th frame of the left disk during the i-th trial and $\mu_L^i$ is the mean of the generative Gaussian distribution for the left disc in the i-th trial. We computed the PKs standard error of the mean using bootstrap with 1000 repetitions.

To compute the PK of the DWM we simulated Eq. 6 using stimuli with the exact same temporal fluctuations in evidence than the stimuli presented to the subjects. We modeled it by updating $\mu^i(t)$ from Eq. 3 with the difference in brightness at each time between the right and left disk:

$$\mu^i(t) = S_R^i(t) - S_L^i(t). \tag{37}$$

Note that in this framework the stimulus fluctuations were set to zero $\sigma_S = 0$ because $\sigma_S$ was captured inside $\mu^i(t)$. The DWM parameters ($\alpha = -0.8$, $\sigma_I = 0.3$, and $\tau = 200$ ms) were tuned to account for the change from primacy to recency with the stimulus duration.

**Primacy-recency index for the two pulses trials**. In Fig. 7, we define the primacy-recency index

$$PRI = \frac{\beta_2 - \beta_1}{\beta_1 + \beta_2} \tag{38}$$

where $\beta_1$ and $\beta_2$ are the coefficients of a logistic regression with the coherence of the first and second pulse as predictors:

$$logit(P_C) = \beta_0 + \beta_1 coh_1 + \beta_2 coh_2 \tag{39}$$

Similar to the Normalized PK slope, the primacy-recency index ranges from −1 (primacy) to 1 (recency).

**DWM fitting**. In Fig. 7, we use data from two studies performing the same experiments[35,40]. We extract the accuracy of the subjects directly from the paper figures (with GraphClick, a software to extract data from graphs) and the number of trials from the methods of the papers. We pool the data from the two experiment and we compute the mean accuracy in each condition $i$ as

$$P_i = \frac{P_i^K N_i^K + P_i^T N_i^T}{N_i^T + N_i^K}, \tag{40}$$

where $N_i$ is the number of trials in condition i, the data with superindex $K$ and $T$ were extracted from[35] and[40] respectively. The 95% confidence interval of $P_i$ is:

$$P_i \pm 1.96 \sqrt{\frac{P_{C,i}(1 - P_{C,i})}{N_i^T + N_i^K}}, \tag{41}$$

In these experiments, the human subjects had to discriminate between left and right motion direction of a random dots stimulus. The experimenters interleaved trials with one and two pulses of 120 ms. For single pulse trials the possible coherence levels were 0, 3.2, 6.4, 12.8, 25.6, and 51.2%. For double pulse trials, the pulses were separated by a delay of 0, 120, 360, or 1080 ms and the coherences were randomly chosen from 3.2, 6.4, and 12.8% (nine different coherence sequences). In both papers, they reported that the subjects' accuracy in double pulses trials was independent of the delay. Thus we assume that, in the DWM, the internal noise was too small to drive transitions during the delay and we pool the data across delays to compute the accuracy for each coherence sequence. We fit the model by maximizing the log-likelihood (Nelder–Mead algorithm):

$$LL = \sum_i^{N_i} N_{C,i} P_i + N_{E,i}(1 - P_i), \tag{42}$$

where $N_{C,i}$ and $N_{E,i}$ are the number of correct and error trials for each coherence sequence i whereas $_{P_i}$ is the accuracy for sequence i predicted by the DWM. For single pulse trials, we computed $P_i$ as

$$P_i^1 = P_0(1 - p_E) + (1 - P_0)p_C, \tag{43}$$

where $P_0$, $p_C$, and $p_E$ were computed using Eqs. 14, 17, and 18 whereas the super

index indicates the pulse number. Note that we are assuming that the time spent for the decision variable in the unstable state is short compared with the pulse duration. With this assumption, the decision variable starts in the correct attractor with probability $P_0$. Similarly for double pulse trials the probability of correct is:

$$P_i^2 = P_C^1(1 - p_E) + (1 - P_C^1)p_C. \tag{44}$$

The potential and the diffusion equation can be written as

$$\varphi(X) = \mu x - \alpha x^2 + x^4 \text{ and} \tag{45}$$

$$\tau \frac{dx}{dt} = -\frac{d\varphi}{dx} + \sigma \xi(t), \tag{46}$$

where $\mu$ is a linear scaling of the coherence to $x$ units ($\mu = kcoh$) and $\sigma$ represents the two sources of noise, the internal noise and the stimulus fluctuations $\sigma = \sqrt{\sigma_I + \sigma_S}$. The two sources of noise cannot be fitted separately because the only difference between them is that the internal noise is also activated during the delay (Fig. 7a). But internal noise does not have any impact during the delay. Thus it is impossible to distinguish $\sigma_I$ in the range $(0, \sigma_I^{max})$ where $\sigma_I^{max}$ is the maximum $\sigma_I$ without transitions during the delay. For this reason, we assume that there are no transitions during the delay and we only fit the total noise $\sigma$. The parameters that maximize Eq. 42 and their 95% confidence interval are $k^* = 0.012 \pm 0.0011$, $\alpha^* = 0.70 \pm 0.05$, $\sigma^* = 0.52 \pm 0.05$, and $\tau^* = 3.3 \pm 0.5$. To compute the confidence intervals, we assume that the likelihood function around the best-fit parameters is a multi-dimensional Gaussian. Then the confidence intervals are two times the diagonal of the inverse of the Hessian matrix[17,65]. The Hessian matrix is the matrix of second derivatives and we compute it numerically using the finite difference method.

Although we cannot fit the internal and the stimulus sources of noise separately, we can study the range of internal noise $(0, \sigma_I^{max})$ that produces a negligible number of transitions (<1%) during the delay (up to 1 s) and thus is compatible with the psychophysical data. For the parameters that maximize the likelihood this range is (0, 0.32), indicating that the DWM is robust to perturbations during the delay even when the magnitude of the internal noise represents a substantial part of the total noise ($\sigma_I^{max}/\sigma_S = 0.8$)(Fig. 7d). We also compute the accuracy of the PI as a function of the delay (Fig. 7d). To be able to compare both models, we adjust the scaling factor of the evidence to match subjects' accuracy for the shortest delay ($\mu_{PI} = 0.44k \times coh$ where $k$ is the scaling of the DWM), and we use the parameters $\tau$, $\sigma_S$, and $\sigma_i$ that maximize the DWM.

**Model for n-choice decision making**. To model a categorization task with $n = 3$ choices (Supplementary Fig. 5) we simulated a system of standard nonlinear coupled rate equations (see e.g., Equation (38) in[66]):

$$\tau \frac{dr_1}{dt} = -r_1 + \phi(sr_1 - cr_I + I_1) + \xi_1(t),$$
$$\tau \frac{dr_2}{dt} = -r_2 + \phi(sr_2 - cr_I + I_2) + \xi_2(t),$$
$$\tau \frac{dr_3}{dt} = -r_3 + \phi(sr_3 - cr_I + I_3) + \xi_3(t),$$
$$\tau \frac{dr_I}{dt} = -r_I + \phi\left(\frac{g}{3}(r_1 + r_2 + r_3) + I_I\right) + \xi_I(t), \tag{47}$$

where $\xi_i$ is a Gaussian white noise process with amplitude $\sigma$ for i = 1,2,3 and amplitude $\sigma_I$ for i = I, and with transfer function $\phi(x) = 0$ for $x < 0$, $\phi(x) = x^2$ for $0 \leq x \leq 1$ and $\phi(x) = 2\sqrt{x - 3/4}$ for $x > 1$. The $n = 4$ case is a simple extension of these equations (the general system of rate equations for $n > 2$ can be found in[66]). The parameters used for the simulations were s = 0.694, c = g = $\sqrt{5}$, $\tau = 20$ ms, $\tau_I = 10$ ms, $I_I = 0$, and $\sigma_I = 0$. For $n = 3$ the inputs were taken as $I_1 = I + 2\Delta I/3$, $I_2 = I - \Delta I/3$, $I_3 = I - \Delta I/3$, while for $n = 4$ they were $I_1 = I + 3\Delta I/4$, $I_2 = I - \Delta I/4$, $I_3 = I - \Delta I/4$, $I_4 = I - \Delta I/4$, with $I = 2.25$ and $\Delta I = 0.03*$s. For the top panels in Supplementary Fig. 5 the values of the noise strength where $\sigma = 0.18$ and $\sigma = 0.13$ for $n = 3$ and 4, respectively. The accuracy was calculated as the fraction of trials ($N = 10,000$) in which the highest firing rate at the end of the trial was $r_1$.

**Reporting Summary**. Further information on research design is available in the Nature Research Reporting Summary linked to this article.

## Data availability
Data shown in Fig. 6 can be accessed here: https://doi.org/10.1371/journal.pcbi.1004667.
The data shown in Fig. 7 was extracted directly from the manuscripts[35,40] using GraphClick.

## Code availability
The codes to simulate the DWM and canonical models and generate the figures of the paper are available at https://bitbucket.org/delaRochaLab/flexible-categorization.
The code and analysis scripts for the spiking neural network simulations are available at https://github.com/wimmerlab/flexcat-spiking.

# ARTICLE

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

## Acknowledgements

We thank Tobias H. Donner and Niklas Wilming for excellent discussions. The research leading to these results has received funding from "la Caixa" Foundation (to G.P.O.), the Spanish Ministry of Economy and Competitiveness together with the European Regional Development Fund (RYC-2015-17236 and BFU2017-86026-R to K.W, MTM2015-71509-C2-1-R and RTI2018-097570-B-I00 to A.R. and SAF2015-70324-R to J.R.) and from the Generalitat de Catalunya (grant AGAUR 2017 SGR 1565 to A.R., J.d.I.R., and K.W.). This work has received funding from the European Research Council (ERC) under the European Union's Horizon 2020 research and innovation program (ERC-2015-CoG - 683209 PRIORS to J.d.I.R.). Part of this work was developed at the building Centre Esther Koplowitz, Barcelona.

## Author contributions

G.P.O., K.W., A.R., and J.d.l.R. contributed to the design of the study and to the interpretation of the results. G.P.O. performed the simulations and the analysis of the canonical and the double well models. G.P.O. and A.R. derived analytical expressions from the DWM. K.W. and G.P.O. performed the simulations and analysis of the spiking network. G.P.O., K.W., A.R., and J.d.l.R. wrote the paper.

## Competing interests

The authors declare no competing interests.
