## [Peer Review File · Nature Communications]

Reviewers' Comments:

Reviewer #1:

Remarks to the Author:

This manuscript reports the existence of a form of temporal evidence weighting that distinguishes attractor and (continuous) integrator models of two-choice decision making. The authors demonstrate that increasing key parameters of the decision-making process, including stimulus fluctuations and durations, can flip attractor models from weighting initial evidence to weighting the most recent evidence, through changing how easily an initial incorrect decision can be reversed. Their key insight is that this flipping contains an intermediate regime where it is easier to move from the incorrect option's basin of attraction to the correct option's basin of attraction than vice-versa. They show that this intermediate regime predicts choice accuracy and choice consistency are non-monotonic functions of stimulus properties, that this regime and non-monotonicity exist in full spiking-network models, and that this intermediate regime can account for experiments showing stimulus-dependent changes in evidence weighting and showing combinations of evidence after a delay. Crucially, these predictions are qualitatively distinct from continuous integrator accounts.

The existence of this intermediate regime is new, as far as I know, and insightful for both behavioural and neural theories of decision making. The manuscript is well-written, and the results clearly presented. As the central result depends on the qualitative asymmetry of the incorrect/correct option attractors, which is well explained, I could not see any major issues here. So I have only minor comments.

Minor comments:

(1) Figure 1: I could not match the PK area and slope summaries (panels g-h) to the example PKs (panel d):

- why does the PK area for the reflecting boundary increase to 1 with increasing stimulus fluctuations in panel g? The examples in panel d suggest the PKs is anti-symmetric to those of the absorbing boundary, and so the area curves should overlap, but do not; moreover, the reflecting boundary creates recency weighting at high stimulus fluctuations, but the normalised PK area approaching 1 suggests this is the same as a perfect integrator?

- the PK slope curves in panel h are inverted: the absorbing model has negative slopes, the reflecting model has positive slopes.

(2) SI Fig 3: here is proposed an outline of an experiment to distinguish the attractor and continuous integrator models. This seems a crucial contribution of the paper, but the proposed task design is not clear to me. Would appreciate clarifying. The key variable seems to be a read-out of the decision-variable after the first pulse of evidence, but this is only accessible through neural recordings. Interpretation of these would be contentious. Given the numerous behavioural predictions made by the intermediate regime, is there not a task design that can distinguish the types of model based on e.g. choice accuracy and consistency combined?

(3) Discussion: would be good to consider generalisation to the multi-alternative case. While $N=2$ options is usual for a lab task, it is not usual for real decision making. Could the authors speculate on if and how their intermediate regime would work for $N>2$ options? For example, presumably it would still exist if the stimulus properties could guarantee that the "correct" option basin of attraction was deeper than all others.

(4) Suggested text changes

- Abstract: "Our findings point to correcting transitions as.." Unclear what "correcting transitions" refers to without reading the paper

- Figure 1, legend for panel b: "inset in e"

- Define Psychophysical Kernels when first mentioned (Fig 1 and main text)

- pg 7 "(Figure 2c, red line)" - there are three red lines in Fig 2c
- Discussion, first paragraph: (Figure 5) -> (Figure 6); (Figure 6) -> (Figure 7)

Mark Humphries

Reviewer #2:

Remarks to the Author:

The authors present a computational and experimental study in which two classical approaches to model decision making, the drift-diffusion model and the attractor model, are analyzed. They find that, only for the attractor model, a novel regime may be found which allows for flexible categorization --giving the model a chance to correct choice mistakes when noise fluctuations or stimulus duration are large enough. This regime shifts the response of the model between primacy and recency effects, and it has interesting implications for choice accuracy and consistency. The authors also perform psychophysical experiments on humans to test their model predictions and find the expected effects for long stimulus duration, and apply their model to explain primacy-recency effects in data on sensory integration during a working memory task.

I consider this work, which combines theoretical calculations, numerical simulations and psychophysical experiments, of a very high quality. I think it could be relevant for the readers of Nature Communications given its breadth and insight on the dynamical regimes of decision making models. I have, however, several concerns which should be properly addressed before further considerations:

- 1) How robust is the flexible categorization phenomenon with respect to the input bias? It seems to me that, for a small bias, fluctuations will affect the basins of both attractors almost equally and the beneficial effect of 'correcting wrong choices' will disappear, while for a large bias fluctuations won't be able to correct a wrong choice. Authors might want to run simulations to test whether flexible categorization holds for a wide range of input values.
- 2) It would be interesting to address whether flexible categorization could also emerge in canonical DDMs with collapsing absorbing bounds, modeling the effect of urgency signals in decision making. Ideally, collapsing bounds could display a PK with a recency effect (if fluctuations are small, so that only when bounds collapse they make a difference) or with a primacy effect (if fluctuations are large, effectively working as a DDM). Depending on how this is done it might lead to the emergence of flexible categorization in these simpler models too.
- 3) What is the point of considering internal and external fluctuations independently? They are both assumed to be Gaussian white noise, therefore their distinction is purely semantic in the model. This makes the otherwise appealing section of choice consistency less interesting. Authors could extend this section by adding realistic differences between internal and external noise in real circuits (for example, assuming that external fluctuations may vary slower than internal ones, thus assuming the external noise as a slow OU process) and explore their implications for the consistency results.
- 4) I find the simulation of the LIF network not convincing in its current state. In particular, the synapse models contain only simplified postsynaptic receptor dynamics (similar to AMPA and GABA), however the time constant is too large for AMPA dynamics (12 ms) and too small for GABA dynamics (1 ms). Models with a more realistic synaptic dynamics, with faster AMPA dynamics and also including NMDA dynamics, are well known in the literature (Wang, Neuron 2002); authors should use these more realistic models to provide a more convincing statement regarding biophysical plausibility.

Minor points:

-Please introduce the concept of psychophysical kernel early on in the paper, as it might about confusion.

-Figure 7 would benefit from an extra panel showing a schematic of the working memory task.

Flexible categorization in perceptual decision making
G. Prat-Ortega , K. Wimmer , A. Roxin and J. de la Rocha

Response to Reviewers

We would like to thank the two reviewers for the very positive comments about our manuscript. We would also like to thank them for all the enquiries they have raised. We believe that in addressing them the manuscript has clearly improved.

Below we address the comments of each referee (**rewritten in blue**), providing explanations and clarifications (**in black**) and pasting the new additions and edits to the paper that address these comments (colored **in red** in this document and the main text).

REVIEWER COMMENTS

Reviewer #1 (Remarks to the Author):

This manuscript reports the existence of a form of temporal evidence weighting that distinguishes attractor and (continuous) integrator models of two-choice decision making. The authors (...). Crucially, these predictions are qualitatively distinct from continuous integrator accounts.

The existence of this intermediate regime is new, as far as I know, and insightful for both behavioural and neural theories of decision making. The manuscript is well-written, and the results clearly presented. As the central result depends on the qualitative asymmetry of the incorrect/correct option attractors, which is well explained, I could not see any major issues here. So I have only minor comments.

Minor comments:

(1) Figure 1: I could not match the PK area and slope summaries (panels g-h) to the example PKs (panel d):

- why does the PK area for the reflecting boundary increase to 1 with increasing stimulus fluctuations in panel g? The examples in panel d suggest the PKs is anti-symmetric to those of the absorbing boundary, and so the area curves should overlap, but do not; moreover, the reflecting boundary creates recency weighting at high stimulus fluctuations, but the normalised PK area approaching 1 suggests this is the same as a perfect integrator?

We thank the reviewer for raising this point. There was a mistake in the color code of the curves in panels g and h of Fig. 1. We have corrected these colors in the new version of the manuscript. As expected, the area of Perfect Integrator increases to 1 whereas the area of the other two drift diffusion models (with absorbing bounds and with reflecting bounds) decreases

to zero (the two curves overlap). As for the slopes of the PK (panel h), Perfect integrator→ constant, DDMreflecting→ increases, DDMabsorbing→ decreases.

- the PK slope curves in panel h are inverted: the absorbing model has negative slopes, the reflecting model has positive slopes.

This is the same error we address in the response to your previous point. It is now corrected.

(2) SI Fig 3: here is proposed an outline of an experiment to distinguish the attractor and continuous integrator models. This seems a crucial contribution of the paper, but the proposed task design is not clear to me. Would appreciate clarifying. The key variable seems to be a read-out of the decision-variable after the first pulse of evidence, but this is only accessible through neural recordings. Interpretation of these would be contentious. Given the numerous behavioural predictions made by the intermediate regime, is there not a task design that can distinguish the types of model based on e.g. choice accuracy and consistency combined?

We have now changed this Figure which is now Fig. 6 to add a more detailed explanation of the experiment we propose. In the same vein, the caption of that figure has been substantially expanded and there is a longer justification in the Discussion for the necessity of such an experiment. The Discussion on this point now reads (p. 23):

Models that assume perfect integration of evidence can generally store a parametric value in short-term memory but they are susceptible to undergoing diffusion over time, causing a drop in memory precision as the delay increases(Wimmer et al. 2014; Compte 2000). In contrast, the fact that the accuracy did not decrease with delay duration suggests that the information stored in memory could be categorical instead of parametric(Fleming et al. 2013; Inagaki et al. 2019), a feature naturally captured by the DWM (Figure 7d). Alternatively, it could reflect a parametric memory with negligible internal noise (Waskom and Kiani 2018). Interpreting neural recordings can also be non-conclusive as different areas can simultaneously represent stimulus information with different levels of categorization (Hanks et al. 2015). To get over these shortcomings in understanding whether the stored information is categorical or parametric, we propose an experiment that combining electrophysiology with psychophysics can qualitatively distinguish between these two alternatives (see Supplementary Figure 6).

The new Supplementary Figure 6 is also pasted here for convenience:

Supplementary Figure 6 | Categorical versus parametric working memory.

The referee also asks for a simpler version of the proposed experiment in which, using only behavioral data (“accuracy and consistency combined”), one may distinguish between the Perfect Integrator (PI) and the Double Well Model (DWM). We had put some thought on this possibility and have now put some more. We can however not think of an unambiguous way to qualitatively distinguish the two models using only the behavioral responses. Here is why: the qualitative difference between the PI and DWM is their ability to maintain a stable representation of the memory during the mnemonic *delay period* in the face of noise: (1) the DWM stores a categorical value, it is more robust to noise and shows a null or weak dependence of accuracy on delay length D . (2) The PI stores a parametric value at the cost of being more susceptible to internal noise and hence shows a decrease of the accuracy as a function of the delay length D . Because the data shows no dependence of the accuracy on D (Fig. 2 in Kiani et al 2013), we can discard the PI model with noise, but still it could be either (1) a DWM + noise or (2) a PI with marginal noise.

Can we assess whether the internal noise is large or marginal? In principle one can use the consistency for that purpose: if the internal noise was marginal, the consistency of the responses to the exact same stimulus should be large. Although there are not many studies reporting consistency, those that have show that it is typically low (e.g. < 70% in Nienborg & Cumming 2009), supporting the idea that the internal noise is relatively large. Although this could in principle provide support against the PI + marginal noise, the most plausible explanation would be that the noise is high during the stimulus presentation but much lower

during the delay. This distinction between sensory noise and accumulation noise has been already addressed in several studies reaching the conclusion that it is the sensory noise associated with the stimulus the dominant source of noise (Brunton et al 2013). This is in agreement with recent work showing marginal noise in the accumulation and maintenance of sensory information (Waskom and Kiani 2018). One would then need to separate the noise during the stimulus presentation and during the delay. But because there is no dependence of the accuracy with the Delay, there would be an indeterminacy in the value of the delay noise (one could only obtain an upper bound) and the comparison between PI and DWM would end up being a quantitative comparison. Our proposed experiment relies on a qualitative difference between PI and DWM and hence we believe that, although technically more challenging (i.e. it requires electrophysiological recordings) it would be more conclusive.

(3) Discussion: would be good to consider generalisation to the multi-alternative case. While $N=2$ options is usual for a lab task, it is not usual for real decision making. Could the authors speculate on if and how their intermediate regime would work for $N>2$ options? For example, presumably it would still exist if the stimulus properties could guarantee that the “correct” option basin of attraction was deeper than all others.

We thank the reviewer for raising this point. One of us has recently published a paper on how to generalize the drift diffusion model for the case of n choices (Roxin, 2019). We have taken advantage of this fact and, extending the analysis of n -choice systems of that work, have explored the generalisation to $n>2$ choices that the reviewer has suggested (new Fig. 6): interestingly, our preliminary evidence suggests that the non-monotonicity of the accuracy versus the magnitude of the stimulus fluctuations at different stimulus durations T persists. Although further analysis of these multiple-choice networks is needed, we have considered that showing these examples for $n=3$ and $n=4$ as a Supplementary Figure increases the breadth of the main result of the paper and hence strengthens our findings.

We have included this extended analysis In the Discussion (p. 22) where we now write:

One interesting question is whether correcting changes-of-mind could generate similar non-linear effects as those reported here (Fig. 3a-b) in tasks with $n>2$ choices. A preliminary analysis using rate-based networks suggests that this is in fact the case (Supplementary Fig. 5). We simulated rate networks composed of n excitatory populations competing with each other via mutual inhibition and found that in the winner-take-all regime, strong stimulus fluctuations causing attractor transitions could have a beneficial effect and yield a non-monotonic psychometric curve $P(\sigma_S)$ (Supplementary Fig. 5e-f). Thus, although a more detailed analysis of these multiple-choice networks is needed, these examples suggest that the asymmetry between correcting and error transitions underlying the raise in accuracy with σ_S , was a general mechanism that may be in play in tasks with more than two choices.

We also paste here the new Supplementary Fig. 5 for the referee's convenience:

Supplementary Figure 5 | Flexible categorization dynamics in a multiple-choice task.

The Methods section has been correspondingly updated to include the description of this n-choice network.

(4) Suggested text changes

- Abstract: "Our findings point to correcting transitions as.." Unclear what "correcting transitions" refers to without reading the paper

We have now changed the wording to "correcting decision reversals" which is clearer because previously in the abstract we talk about "...fluctuations are strong enough to **reverse** initial categorizations, but only if they are incorrect. This asymmetry in the **reversing** probability, ...".

- Figure 1, legend for panel b: “inset in e”

Corrected, thank you.

- Define Psychophysical Kernels when first mentioned (Fig 1 and main text)

We have added the following sentence in the main text (pag 5):

To study the impact of these bounds, we computed the Psychophysical Kernel (PK) which measures the impact of the stimulus fluctuations during the course of the stimulus (see Methods).

And also in the caption from Fig. 1 we have added:

The PK measures the time-resolved impact of the stimulus fluctuations on choice (see Methods).

- pg 7 “(Figure 2c, red line)” - there are three red lines in Fig 2c

We have now added (text in light red is new):

(Figure 2c, red line, third from the left)

- Discussion, first paragraph: (Figure 5) -> (Figure 6); (Figure 6) -> (Figure 7)

Thanks for pointing this out. It has now been solved.

Reviewer #2 (Remarks to the Author):

The authors present a computational and experimental study in which two classical approaches to model decision making, the drift-diffusion model and the attractor model, are analyzed. They find that, only for the attractor model, a novel regime may be found which allows for flexible categorization (...)

I consider this work, which combines theoretical calculations, numerical simulations and psychophysical experiments, of a very high quality. I think it could be relevant for the readers of Nature Communications given its breadth and insight on the dynamical regimes of decision making models. I have, however, several concerns which should be properly addressed before further considerations:

1) How robust is the flexible categorization phenomenon with respect to the input bias? It seems to me that, for a small bias, fluctuations will affect the basins of both attractors almost equally and the beneficial effect of 'correcting wrong choices' will disappear, while for a large bias fluctuations won't be able to correct a wrong choice. Authors might want to run simulations to test whether flexible categorization holds for a wide range of input values.

The intuition of the referee is correct on this point. Near zero and very large values of the mean stimulus evidence μ are conditions in which the non-monotonicity goes away. However, for all intermediate values the effect is very clear. This is an aspect that we had addressed in our original submission but that the referee may have missed. We have now made this point clearer in the main text. On page 11 we had written (additions in light red):

We next asked whether the non-monotonicity of the psychometric curve was robust to variation of other parameters such as the mean stimulus evidence μ , the stimulus duration T and the internal noise σ_i . We found that the non-monotonicity was robustly obtained over a broad range of μ , ranging from small values just above zero to a critical value beyond which the curve became monotonically decreasing (see Eq. 41 for the critical value and Supplementary Figure 3).

We have now added a reference to the Equation 41 in the Supplementary Information where we compute the critical value of μ_C above which the accuracy decreases monotonically

This question had been directly explored in Supplementary Fig. 3 which we paste again here:

Supplementary Figure 3 | Critical internal noise and mean stimulus evidence compatible with the non-monotonic relation between accuracy and stimulus fluctuations.

To make the conclusion of the analysis clear, we had written at end of the paragraph in Page 11:

(...) In sum, the non-monotonicity of the psychometric curve was a robust effect which did not require fine-tuning of the mean stimulus evidence μ or stimulus duration T . It was most prominent for μ values yielding an intermediate accuracy (i.e. $P \sim 0.75$), long stimulus durations and weak internal noise.

2) It would be interesting to address whether flexible categorization could also emerge in canonical DDMs with collapsing absorbing bounds, modeling the effect of urgency signals in decision making. Ideally, collapsing bounds could display a PK with a recency effect (if fluctuations are small, so that only when bounds collapse they make a difference) or with a primacy effect (if fluctuations are large, effectively working as a DDM). Depending on how this is done it might lead to the emergence of flexible categorization in these simpler models too.

We thank the referee for this interesting proposal. We have now simulated the drift diffusion model with collapsing bounds: increasing the magnitude of the stimulus fluctuations (σ_s) in the DDMA with collapsing bounds did not replicate the crossover from primacy to recency found in the Double Well Model. We also did not find a regime similar to the flexible categorization. Because the bounds can be reached earlier when they collapse, the integration becomes more

transient slightly increasing the primacy degree of the DDMA. This result has been included in the main text and in a Supplementary Figure 1:

Including collapsing bounds in the DDMA did not modify qualitatively the picture, with the integration becoming more transient as the velocity of the collapsing bounds increases (Supplementary Fig. 1).

We paste here the Supplementary Fig. 1 for convenience:

Supplementary Figure 1 | Dynamics of evidence accumulation in the drift diffusion model with collapsing absorbing bounds.

3) What is the point of considering internal and external fluctuations independently? They are both assumed to be Gaussian white noise, therefore their distinction is purely semantic in the model. This makes the otherwise appealing section of choice consistency less interesting.

Authors could extend this section by adding realistic differences between internal and external noise in real circuits (for example, assuming that external fluctuations may vary slower than internal ones, thus assuming the external noise as a slow OU process) and explore their implications for the consistency results.

In our manuscript, we initially assume that the noise is white for mathematical tractability: our analytical expressions for the accuracy (eqs. 13-19) are based on Kramer's theory which assumes white noise. The main results of the paper, namely the cross-over from primacy to recency and the non-monotonic psychometric curve, do not however depend on this assumption. In the paper we show two examples in which the noise is not white:

- In the spiking network with current-based synapses, the external noise is an OU process with time constant $\tau=20$ ms. Moreover, in this network the internal noise is also not white as it is generated by the stochastic dynamics of the network plus the synaptic time constants which are finite. In the newly added spiking network with conductance-based synapses (see response to comment 4 of reviewer 2), we used a larger time constant for the external noise ($\tau = 100$ ms). In both cases, the main results could be easily reproduced with these colored noises.
- In the model fitted to the psychophysical data from Bronfman et al. 2016, we assumed that the external fluctuations are structured in 100 ms frames. That is, they mimic the temporal structure of the visual stimulus in the experiment. The main results could also be reproduced with this temporal structure external fluctuations.

We realize that this generalization of the main results about the time-scale of the noise was not explicitly mentioned in our original submission. We have thus added a few sentences in the Results to remark that the results do not depend on the noise being white.

On page 15 in the Results:

Notice that in contrast with the DWM in which the noise was white (i.e. temporally uncorrelated), in this network the external noise was colored (stimulus was an Ornstein-Uhlenbeck process with $\tau_1 = 20$ ms) and the internal fluctuations reflected the stochasticity of the spiking network dynamics which are strongly affected by the synaptic time scales.

And also in page 15:

Thus, the signatures of attractor dynamics that we had identified **did not depend on the simplifying assumptions of the DWM** and could be replicated in an attractor network with more biophysically plausible parameters.

In caption of Fig. 6:

(c-f) Example traces of the decision variable of the fitted DWM (c,e) and the stimulus (d,f) for 1 and 3 s trials. Notice that the stimulus fluctuations mimicked the visual stimulus which was made of time frames of 100 ms.

Regarding the difference between internal and external noises, in our theory which assumes both to be white, they are mathematically identical. Conceptually however, they are very different because one is controlled by the experimenter whereas the other is not. We think that making this difference is very important. In particular, the distinction between the two has allowed us to talk about consistency and unveil a non-monotonicity in the consistency as a function of the magnitude of the stimulus fluctuations (Fig. 4). If the noise was only external, consistency would always be one. If the noise was only internal, then we could not make any of the analysis of the manuscript in which we systematically vary the magnitude of the stimulus fluctuations. Moreover, recognizing that a fraction of the noise is inaccessible (internal), is important when varying the magnitude of the external fluctuations as there are regimes of the system which you cannot reach no matter how small your stimulus fluctuations are. That is exactly the point we make in Figure 1f : the non-monotonicity of the accuracy vs σ_s may be impossible to reveal if the internal noise σ_I is too large.

4) I find the simulation of the LIF network not convincing in its current state. In particular, the synapse models contain only simplified postsynaptic receptor dynamics (similar to AMPA and GABA), however the time constant is too large for AMPA dynamics (12 ms) and too small for GABA dynamics (1 ms). Models with a more realistic synaptic dynamics, with faster AMPA dynamics and also including NMDA dynamics, are well known in the literature (Wang, Neuron 2002); authors should use these more realistic models to provide a more convincing statement regarding biophysical plausibility.

We thank the reviewer for raising this point. We acknowledge that although the spiking network we had included in original submission was clearly more “biophysically realistic” than the one dimensional DWM, it still lacked important biophysical aspects such as realistic synaptic conductances. We have now run simulations using the exact network presented in the classic paper by Wang (2002) which is made of leaky integrate-and-fire neurons with conductance based synapses (instead of current based synapses as our original network) and contains AMPA, NMDA and GABA synapses with realistic times scales (all parameters are taken from Wang 2002). The results are qualitatively the same and the only change is that, because the dynamics of the network are slower, the stimulus durations required to reveal the non-monotonicity of the psychometric curve are larger. We think that this slowing is not a direct consequence of the more realism of the network but it is due to the fact that this network was originally tuned to be able to integrate stimuli over long time scales (i.e. it was designed to be slow).

We have added the results of this new network in a new Supplementary Figure 4:
In the results (page 15) we now refer to this new network:

We went one step further in including biophysical detail and confirmed that a conductance-based spiking neural network model with explicit AMPA, GABA and NMDA receptor dynamics²⁰ showed qualitatively the same behavior (Supplementary Figure 4).

We paste here the new Supplementary Figure for the referee's convenience:

Supplementary Figure 4 | Flexible categorization dynamics in a spiking neural network model with AMPA, GABA and NMDA receptor dynamics.

Minor points:

-Please introduce the concept of psychophysical kernel early on in the paper, as it might about confusion.

We have added the following sentence in the main text (pag 5):

To quantify this impact, we used the Psychophysical Kernel (PK) which measures the influence of the stimulus fluctuations on the decision during the course of the stimulus (see Methods):

-Figure 7 would benefit from an extra panel showing a schematic of the working memory task.

We have now added a schema of the task in figure 7 panel a which illustrates the task scheme. We paste here the figure with the new panel for the referee's convenience:

Figure 7 | The flexible categorization regime accounts for the combination of two pulses of evidence during a working memory task

Reviewers' Comments:

Reviewer #1:

Remarks to the Author:

The authors have done an excellent job of fully addressing my relatively minor comments. I particularly appreciated that they went beyond a discussion of the $N > 2$ case to simulating a model (Supplemental Figure 5), which suggests the asymmetric changes-of-mind, and hence the non-monotonic psychometric curves, still exist for more than two decisions.

I think the paper is suitable for publication. I have just a handful of suggested text changes for further clarity

- Methods, pg38, after Eq 61: the transfer function is given as ϕ in the text, but not in Eq 61.
- Supplementary Figure 6:
 - it took me a while to realise the basic idea, perhaps a redraft would help. As far as I can tell, the basic idea is to train a classifier to decode the intended decision during the delay using neural data from the single-pulse training trials; then to use the classifier during the delay period of the two-pulse trials to classify the apparently intended decision before the second pulse is delivered. In that way, all the trials of the same intended decision (e.g. right) can be grouped to analyse the effect of the second pulse upon them: if the effect is the same then this is evidence that the decision variable was indeed starting from the same attractor.
 - perhaps use "delay" throughout rather than "WM", as the former is a property of the task, whereas the latter is under volitional control
 - the proposed experiment contrasts the attractor and perfect integrator models, but the main claim of the paper is that the attractor model is distinguishable from the drift diffusion model. Perhaps the authors would like to comment here on how the perfect integrator predictions on this task extend to the drift diffusion model.

Mark Humphries

Reviewer #2:

Remarks to the Author:

After reviewing the changes made by the authors, which include new simulations to address my previous concerns, I am now fully convinced of the results of the manuscript and I'm happy to recommend its publication.

Best regards,

Jorge Mejias

Flexible categorization in perceptual decision making
G. Prat-Ortega , K. Wimmer , A. Roxin and J. de la Rocha

Response to Reviewers

We would like to thank again the two reviewers for the very positive comments about our manuscript.

Below we address the final comments of the first referee (**rewritten in blue**), with our responses **in black**:

REVIEWER COMMENTS

Reviewer #1 (Remarks to the Author):

The authors have done an excellent job of fully addressing my relatively minor comments. I particularly appreciated that they went beyond a discussion of the $N > 2$ case to simulating a model (Supplemental Figure 5), which suggests the asymmetric changes-of-mind, and hence the non-monotonic psychometric curves, still exist for more than two decisions.

We thank the referee for this request. We think that including this new analysis has improved the breadth of our results. The truth is, we were also very curious to know what would happen in the case of $N > 2$, a scenario we had discussed about in the past, but never got around to investigating it. So the request of the referee was the perfect excuse to sit down and test it.

I think the paper is suitable for publication. I have just a handful of suggested text changes for further clarity

- Methods, pg38, after Eq 61: the transfer function is given as ϕ in the text, but not in Eq 61.

Corrected. Thank you.

- Supplementary Figure 6:

- it took me a while to realise the basic idea, perhaps a redraft would help. As far as I can tell, the basic idea is to train a classifier to decode the intended decision during the delay using neural data from the single-pulse training trials; then to use the classifier during the delay period of the two-pulse trials to classify the apparently intended decision before the second pulse is delivered. In that way, all the trials of the same intended decision (e.g. right) can be grouped to analyse the effect of the second pulse upon them: if the effect is the same then this is evidence that the decision variable was indeed starting from the same attractor.

This is correct. That is the idea we are trying to illustrate with this proposed experiment. We have now revised the text explaining this figure and we think it should be clearer now.

- perhaps use “delay” throughout rather than “WM”, as the former is a property of the task, whereas the latter is under volitional control

Thanks for the suggestion. We have changed this now.

- the proposed experiment contrasts the attractor and perfect integrator models, but the main claim of the paper is that the attractor model is distinguishable from the drift diffusion model. Perhaps the authors would like to comment here on how the perfect integrator predictions on this task extend to the drift diffusion model.

The focus of the paper is the comparison of the DWM with the canonical models, namely the absorbing-DDM, the reflecting-DDM and the Perfect Integrator (Fig. 1). However, in the WM task described in Fig. 7 and Supplementary Fig. 6 the comparison is done only against the PI. The reason is that in the literature of WM, the debate is not about whether the DDM or the DWM can explain the data. The Perfect integrator is the default model for WM (see e.g. (Kiani et al J. Neurosci. 2008). It is not clear what the DDM with absorbing bounds would do during the delay period. If the 1st pulse does not make the decision variable reach the one of the two bounds, then it behaves like the PI, whereas if it does, it just ignores the 2nd pulse altogether. In either case it does not seem like a model that could explain this experiment more accurately than the DWM or the PI and little insight would be gained by including it. It is for that reason that we decided to leave the DMM out of the comparison in the WM task.